# GAGA: Gaussianity-Aware Gaussian Approximation for Efficient 3D Molecular Generation

**Jingxiang Qu[1], Wenhan Gao[2], Ruichen Xu[2], Yi Liu[2,1]\***
[1] Department of Computer Science, Stony Brook University
[2] Department of Applied Mathematics and Statistics, Stony Brook University

## Abstract

Gaussian Probability Path based Generative Models (GPPGMs) generate data by reversing a stochastic process that progressively corrupts samples with Gaussian noise. Despite state-of-the-art results in 3D molecular generation, their deployment is hindered by the high cost of long generative trajectories, often requiring hundreds to thousands of steps during training and sampling. In this work, we propose a principled method, named GAGA, to improve generation efficiency without sacrificing training granularity or inference fidelity of GPPGMs. Our key insight is that different data modalities obtain sufficient Gaussianity at markedly different steps during the forward process. Based on this observation, we analytically identify a characteristic step at which molecular data attains sufficient Gaussianity, after which the trajectory can be replaced by a closed-form Gaussian approximation. Unlike existing accelerators that coarsen or reformulate trajectories, our approach preserves full-resolution learning dynamics while avoiding redundant transport through truncated distributional states. Experiments on 3D molecular generation benchmarks demonstrate that our GAGA achieves substantial improvement on both generation quality and computational efficiency. The code is available at `https://github.com/QuJX/GAGA`.

## 1 Introduction

Gaussian Probability Path based Generative Models (GPPGMs) refer to a family of generative models which define a probability path that smoothly transforms a simple Gaussian distribution into the target data distribution (Lipman et al., 2022). Recently, they have demonstrated impressive performance across diverse domains such as images (Li et al., 2019), text (Austin et al., 2021), and molecules (Zhang et al., 2023). However, the generative trajectories are typically modeled as solutions to a stochastic differential equation (SDE) or ordinary differential equation (ODE); such solutions are often discretized by hundreds to thousands of steps for better learning granularity. The heavy computational demand thus becomes one of their key limitations, especially for 3D molecular generation. To improve the efficiency, prior work has largely focused on sampling acceleration, for example, coarsening trajectories with reduced-step solvers (Song et al., 2020; Lu et al., 2022; Karras et al., 2022) and retrieval-based methods (Zhang et al., 2025). While effective for inference, these approaches either compromise trajectory granularity or leave training costs unaffected. Efforts closer to training, such as adaptive priors (Lee et al., 2021; Vignac et al., 2022) and leapfrog initializers for trajectory prediction (Mao et al., 2023), still depend on modifications of the noising process or specialized architectures, rendering them domain-specific and difficult to apply to 3D molecular generation.

In this work, we propose a novel method that improves both training and sampling efficiency of GPPGMs via Gaussian Approximation (GA). Motivated by the translational invariance of molecular data, we leverage this unique property to design GAGA, which operates directly on zero-mean invariant data without any loss of structural information, as analyzed in Sec.2.2. Rather than coarsening the generative trajectories or modifying the predefined noise schedule, our method identifies

---

\*Correspondence to yi.liu.4@stonybrook.edu

a characteristic time step $T^*$ at which the input data distribution has effectively obtained sufficient Gaussianity. *Based on this point, the generative trajectory can be truncated, and the final distribution can be approximated by a tractable Gaussian reference distribution with analytically derived mean and variance, as shown in Fig. 1.* This design yields two key merits absent in existing GPPGMs acceleration methods: **(1) the ability for training acceleration via eliminating ineffective optimization on over-noised inputs**, and **(2) sampling fidelity improvement by maintaining the granularity of the original generative trajectories and focusing more on steps with adequate data information guidance during learning.** We empirically validate our method across different 3D molecular datasets, demonstrating significant improvements in both sampling and training efficiency with high-quality molecular generation.

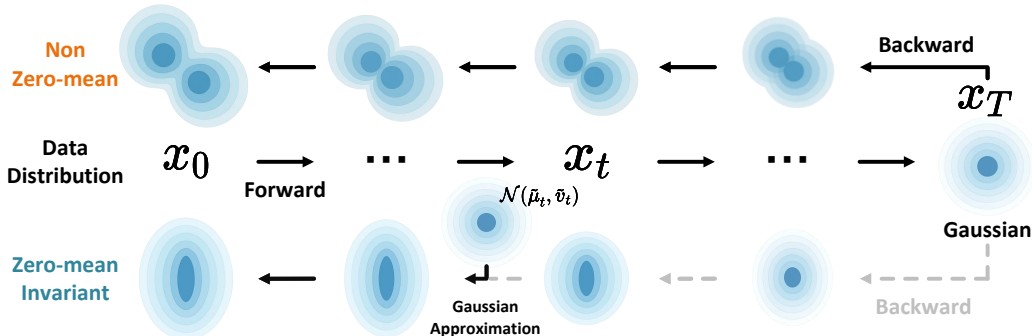

Figure 1: The flowchart of the GAGA, where the forward process is discretized to $T$ steps. In such a case, when the noised data distribution $\boldsymbol{x}_t$ has attained sufficient Gaussianity at timestep $t$, we approximate it with a reference Gaussian $\mathcal{N}(\tilde{\mu}_t, \tilde{v}_t)$. Therefore, the length of the generative trajectory can be reduced from $T$ steps to $t$ steps.

## 2 PRELIMINARIES

### 2.1 GAUSSIAN PROBABILITY PATH BASED GENERATIVE MODELS

GPPGMs define a probability path $q(\boldsymbol{x}_t)$, $t \in [0, 1]$[1], where the trajectory smoothly transforms a simple Gaussian prior into the target data distribution. At $t = 0$, $q(\boldsymbol{x}_0)$ is the data distribution; at $t = 1$, $q(\boldsymbol{x}_1)$ is a standard Gaussian. This probability path perspective gives a continuous interpolation between noise and data. Specifically, these models include the diffusion models (Ho et al., 2020; Song et al., 2021; 2020) and the Gaussian flow matching models (Lipman et al., 2022; Gao et al., 2025)[2]. While the probability path captures the evolution of the marginal distributions $q(\boldsymbol{x}_t), t \in [0, 1]$, a more concrete characterization arises from the conditional probability path $q(\boldsymbol{x}_t \mid \boldsymbol{x}_0), t \in [0, 1]$, which traces the transformation of individual data samples. This path is defined as

$$q(\boldsymbol{x}_t \mid \boldsymbol{x}_0) = \mathcal{N}(\boldsymbol{x}_t \mid \sqrt{\bar{\alpha}_t}\boldsymbol{x}_0, \ \bar{\sigma}_t^2\mathbf{I}), \tag{1}$$

where $\bar{\alpha}_t \in [0, 1]$ with $\bar{\alpha}_0 = 1$ controls the decay of the signal power over time and $\bar{\sigma}_t = \sqrt{1 - \bar{\alpha}_t}$ is the level of injected noise. Typically, $\bar{\alpha}_t$ is monotonically decreasing so that $\bar{\alpha}_1 \approx 0$, and $x_1$ is very close to the pure Gaussian distribution. This conditional Gaussian probability path is used in denoising diffusion models, and it can be shown that the flow trajectories in Gaussian flow matching models also follow the conditional Gaussian probability path (Gao et al., 2025; Ma et al., 2024). Without loss of generality, throughout this paper we assume that the forward process of GPPGMs is variance-preserving (VP). Similar conclusions can be readily extended to other forward settings.

**Learning the Reverse Process**   While the forward conditional path in equation 1 describes the forward noising process from data to Gaussian noise, generative modeling requires the reverse path,

---

[1]We follow the convention in prior works, which simplify $q_t(\boldsymbol{x}_t)$ to $q(\boldsymbol{x}_t)$. In practice, the continuous time interval is discretized into $T$ steps; we use $t \in \{0, 1, ..T\}$ when referring to the discretized implementation.

[2]In this work, we primarily focus on denoising diffusion models and Gaussian flow matching models, which represent the two mainstream frameworks within GPPGMs.

which maps Gaussian samples back to the data distribution. Given equation 1, we can derive a closed-form expression for the reverse process:

$$p\left(x_{t-\Delta t} \mid x_t, x_0\right) = \mathcal{N}\left(x_{t-\Delta t} \mid \mu_t\left(x_t, x_0\right), \tilde{\sigma}_t^2 I\right),$$ (2)

where

$$\mu_t\left(x_t, x_0\right) = \frac{\sqrt{\bar{\alpha}_{t-\Delta t}}\left(1 - \alpha_t\right)}{1 - \bar{\alpha}_t} x_0 + \frac{\sqrt{\alpha_t}\left(1 - \bar{\alpha}_{t-\Delta t}\right)}{1 - \bar{\alpha}_t} x_t,$$

$$\tilde{\sigma}_t^2 = \frac{\left(1 - \alpha_t\right)\left(1 - \bar{\alpha}_{t-\Delta t}\right)}{1 - \bar{\alpha}_t},$$

where the $\alpha_t$ is the discretization step–wise coefficient, defined by $\alpha_t = \frac{\bar{\alpha}_t}{\bar{\alpha}_{t-\Delta t}}, \bar{\alpha}_t = \prod_{s=1}^{t} \alpha_s$. Consequently, the $x_0$ is typically unavailable in the reverse process. Therefore, the objective of GPPGMs is to approximate the reverse process with a parameterized model $\theta$ that only conditions on the observable noisy sample: $p_\theta\left(x_{t-\Delta t} \mid x_t\right) \approx p\left(x_{t-\Delta t} \mid x_t, x_0\right)$. To accomplish this, for diffusion models, we minimize the KL divergence between them: $\text{KL}(p\left(x_{t-\Delta t} \mid x_t, x_0\right) \| p_\theta\left(x_{t-\Delta t} \mid x_t\right))$; since both are assumed to be Gaussian, the KL divergence can be simplified to MSE between the means:

$$\mathcal{L}(\theta) = \mathbb{E}_q\left[\frac{1}{2\sigma_t^2} \left\| \mu_t^\theta(\boldsymbol{x}_t) - \mu_t(x_t, x_0) \right\|^2\right],$$ (3)

For Gaussian flow matching models, the training objective is derived from an ODE perspective; nevertheless, it can be proven that there is an equivalence for both the training objectives and the learned models between diffusion and flow models (Gao et al., 2025), assuming a suitable noise schedule. As a result, the GA techniques and mathematical insights provided in this work can be effectively applied to both diffusion models and Gaussian flow matching models, as demonstrated by the experimental results in Sec. 4.2.

## 2.2 ZERO-MEAN INVARIANCE

A data modality is *zero-mean invariant* if centering each sample by subtracting its empirical mean preserves all the information necessary for downstream modeling. Formally, let $x \in \mathbb{R}^d$ denote a data sample, and define its centered version as:

$$\tilde{\boldsymbol{x}} = \boldsymbol{x} - \frac{1}{d} \sum_{i=1}^{d} \boldsymbol{x}_i \cdot \mathbf{1}_d,$$ (4)

where $\mathbf{1}_d \in \mathbb{R}^d$ is the vector of all 1-s. A data modality is said to satisfy zero-mean invariance if, for all $\boldsymbol{x}$ in the support of the data distribution $p(\boldsymbol{x})$, the transformation $\boldsymbol{x} \mapsto \tilde{\boldsymbol{x}}$ retains the semantic or structural information of the original input.

This property is common in domains where only internal relationships among dimensions carry information, while global offsets are irrelevant or redundant. Typical examples include any representations defined up to an affine baseline or possessing a shift-symmetric structure, such as configurations invariant to global alignment, label encodings invariant to additive bias, or features embedded in contrastive spaces. We provide some detailed examples and the corresponding analysis in Appendix B. Zero-mean invariance permits generative models to operate in a reduced subspace orthogonal to the mean direction, eliminating redundant degrees of freedom. In 3D molecular data, zero-mean invariance is widely employed due to its translational invariance (Hoogeboom et al., 2022; Hong et al., 2025; Xu et al., 2023).

## 3 GAUSSIANITY-AWARE GAUSSIAN APPROXIMATION

Building on the preliminaries, we now introduce our framework for shortening the generative trajectory in GPPGMs via truncation. Rather than executing the full generative trajectories, we identify a characteristic timestep $T^*$ at which the data effectively exhibits sufficient Gaussianity[3]. This enables an analytic truncation, whereby the redundant trajectory is replaced with a direct Gaussian approximation (GA). It significantly improves both computational efficiency and generative fidelity by

---

[3]Following (Berg et al., 2012), in our work, Gaussianity refers to a dataset or random variable distributed according to a Gaussian distribution.

enabling the GPPGM to skip the optimization on over-noised generation trajectory segments while focusing more on the segments with adequate data information guidance in the learning process. It is noted that since both diffusion and flow-matching models share the same Gaussian probability path, as introduced in Sec. 2.1, GA is equivalently applicable to both for trajectory truncation. For simplicity, we follow the setting of diffusion model (Hoogeboom et al., 2022) on the forward process and present our method in this context.

## 3.1 GAUSSIAN APPROXIMATION

GA is commonly employed in statistics to represent intractable conditional or marginal distributions (Berry, 1941; Deng & Zhang, 2020; Chernozhukov et al., 2013). This modeling choice facilitates closed-form expressions for critical quantities, including transition densities, posterior distributions, and variational bounds, which are essential for both optimization and sampling procedures. In GPPGMs, the forward process can be interpreted as progressively pushing the data toward a Gaussian distribution. As illustrated in equation 1, the marginal and transition densities of the trajectories at any finite time index remain Gaussian. However, the intractability of data distribution prevents us from directly calculating the mean and variance of the Gaussian at intermediate timesteps.

Meanwhile, for data modalities that are *zero-mean invariant*, such as molecular coordinates, point clouds, or categorical embeddings, the difficulty of estimating mean can be avoided by enforcing zero-centering as a preprocessing step. Such centering preserves structural information and symmetries (e.g., translational invariance) (Hoogeboom et al., 2022), while consistently ensuring the mean equals to 0, as analyzed in Appendix B.

In addition, the variance remains intractable to obtain exactly. In this paper, we estimate it through the per-sample statistics. Given a dataset $\mathcal{D} = \{\boldsymbol{x}^{(i)}\}_{i=1}^N$ with $\boldsymbol{x}^{(i)} \in \mathbb{R}^d$, we compute

$$v^{(i)} = \frac{1}{d-1} \sum_{j=1}^d (\boldsymbol{x}_j^{(i)} - \mu^{(i)})^2, \quad \text{where} \quad \mu^{(i)} = \frac{1}{d} \sum_{j=1}^d \boldsymbol{x}_j^{(i)}, \tag{5}$$

and aggregate across the dataset to obtain the *average per-sample variance* as $\hat{v} = \frac{1}{N} \sum_{i=1}^N v^{(i)}$. This estimator is unbiased under mild moment conditions (Vershynin, 2012), and we empirically verify that it's an effective estimator for GA in GPPGMs. Therefore, under the VP forward process on zero-meaned data, we plug in $\hat{v}$ into Equation 1 as the variance of data samples, then the mean $\tilde{\mu}_t$ and variance $\tilde{v}_t$ of noised data at intermediate timesteps $t$ can have the following analytic form:

$$\tilde{\mu}_t = \boldsymbol{0}, \quad \tilde{v}_t = 1 - \bar{\alpha}_t(1 - \hat{v}). \tag{6}$$

Consequently, for zero-meaned data, once sufficient noise has been injected at timestep $T^*$, the marginal distribution of $\boldsymbol{x}_{T^*}$ can be approximated by $\mathcal{N}(\boldsymbol{0}, \tilde{v}_{T^*} \mathbf{I})$. This serves as the foundation for our strategy of trajectory truncation.

## 3.2 GAUSSIAN APPROXIMATION AND INITIAL DATA DISTRIBUTION

The analysis above shows that, once sufficient noise is injected, the forward process admits a tractable GA. Nevertheless, the following question arises:

> **(Q)** How do we determine $T^*$ at which the injected noise becomes sufficient for this approximation? Is it the same across different tasks?

To answer this question, we first present Proposition 3.1 to show that $T^*$ is related to properties of the initial data distribution.

**Proposition 3.1.** *Given $t \in [0, T)$ and $K \geq 3$, and the Gaussianity evaluation functional*

$$\mathcal{H}^{(K)}(x) := \beta \big\| \Pi_{D^\perp}(\text{Cov}(x)) \big\|_F + \mathbf{1}_{\{K \geq 3\}} \sum_{k=3}^K w_k \big\| C^{(k)}(x) \big\|_F. \tag{7}$$

*where $\mathbf{1}_{\{\cdot\}}$ is the indicator function, $\beta > 0$ and $w_k > 0$ ($k \geq 3$). $D := \{\text{Diag}(v) : v \in \mathbb{R}^d\}$ is the diagonal subspace and*

$$\Pi_D(\Sigma) := \text{Diag}(\text{diag}\,\Sigma), \quad \Pi_{D^\perp}(\Sigma) := \Sigma - \Pi_D(\Sigma). \tag{8}$$

are the orthogonal projections. $Cov(\cdot)$ and $C^{(k)}(X)$ are the covariance calculator and the $k$-th cumulant tensor, respectively. Let $A, B$ be two initial data distribution, where

$$\mathcal{H}^{(m)}(\boldsymbol{x}_t^A) \ \leq \ \mathcal{H}^{(m)}(\boldsymbol{x}_t^B) \quad \text{for all } m = 2, 3, \ldots, K \tag{9}$$

holds with at least one strict inequality. Then for every $s > t$,

$$\mathcal{H}^{(K)}(\boldsymbol{x}_s^A) \ < \ \mathcal{H}^{(K)}(\boldsymbol{x}_s^B).$$

Consequently, for every $\epsilon > 0$,

$$T_A^* = \inf\{s > t : \mathcal{H}^{(K)}(\boldsymbol{x}_s^A) \leq \epsilon\} \ < \ \inf\{s > t : \mathcal{H}^{(K)}(\boldsymbol{x}_s^B) \leq \epsilon\} = T_B^*. \tag{10}$$

A formal proof is provided in Appendix C. **This proposition establishes that if the initial data distribution is inherently closer to Gaussian, then the corrupted samples achieve sufficient Gaussianity earlier, and the corresponding GA timestep $T^*$ can be smaller, as qualitatively shown in Fig. 2.** In particular, sparse molecular coordinates around equilibrium are closer to Gaussian (Frenkel & Smit, 2023), as empirically quantified in Appendix D. In such case, the GA of molecular data can start at a smaller $T^*$. Since different initial data distribution induces different GA timesteps, we need a principled way to identify the precise $T^*$. Therefore, in Sec. 3.3, we develop a statistical Gaussianity evaluator that serves as an operational test, combining dependency measurement and distributional similarity criteria to precisely identify the GA timestep $T^*$.

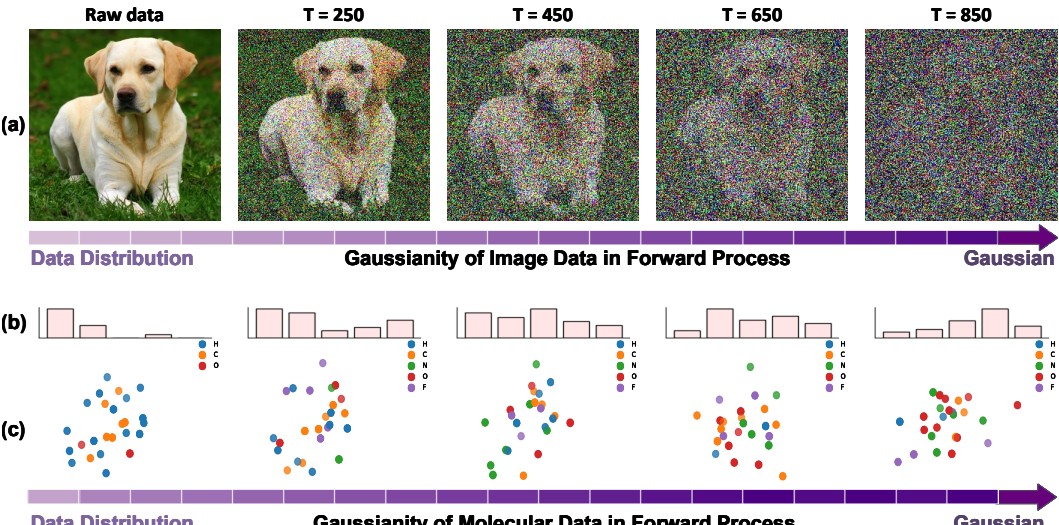

Figure 2: Comparisons of the forward noising process across different data modalities. (a) shows a continuous-valued image matrix, while (b) and (c) illustrate the distribution of molecular data consisting of one-hot vectors for atom types and 3D Euclidean coordinates for atom positions, respectively. **The same schedule for gaussian probability path is applied across all modalities, with the number of forward timesteps $T$ up to 1000.** Despite identical signal-to-noise ratios, molecular data obtains sufficient Gaussianity for significantly fewer steps compared to image data. This comes from the different Gaussianity across initial data distributions, as empirically quantified in Appendix D.

### 3.3 Evaluating Gaussianity: Data Dependency and Distributional Similarity

While the preceding analysis suggests that $\boldsymbol{x}_t$ may be approximated by a Gaussian, the validity of this approximation fundamentally depends on whether the $\boldsymbol{x}_{T^*}$ has gained sufficient Gaussianity for GA. In this section, we present the Gaussianity evaluation method from the perspectives of data dependency and distributional similarity.

**Data Dependency Decay.** The timestep at which data loses its structural information under progressive noise perturbation is critical for establishing the valid Gaussian approximation. Since the Gaussian distribution in GA is independent, the disappearance of data dependency in $\boldsymbol{x}_t$ is a sufficient condition to the independent Gaussian approximation. As shown in Fig. 2, under the same

noise schedule, the timestep at which the data has obtained sufficient Gaussianity strongly depends on the underlying data modality. 3D molecular data combines one-hot atom types and 3D Euclidean coordinates (Liu et al., 2022; Wang et al., 2022; Yan et al., 2022; Wang et al., 2023; Lin et al., 2023; Subedi et al., 2024; Liu et al., 2025; Qu et al., 2025). Due to sparsity and low dimensionality, the initial molecular data distribution preserves less dependency. In contrast, natural images are high-dimensional and spatially correlated, with smooth local structures that preserve structural dependency over many more noising steps. Monitoring the decay of data dependency thus provides a principled criterion for determining the characteristic timestep $T^*$ at which Gaussian approximation becomes valid.

Consequently, to quantify the Gaussianity from the perspective of data dependency, we adopt the mutual information (MI) test (Kraskov et al., 2004) as our evaluation functional. Because exact independence occurs only at the terminal prior $\boldsymbol{x}_1 \sim \mathcal{N}(\mathbf{0}, \mathbf{I})$, we adopt a tolerance $\varepsilon_{\text{dep}} > 0$ and define the dependency-loss timestep as

$$T_{\text{ID}} := \min \left\{ t \,\Big|\, \text{Dep}(\boldsymbol{x}_t) \leq \varepsilon_{\text{dep}} \right\}, \tag{11}$$

where $\text{Dep}(\cdot)$ denotes the MI-based dependency evaluator. Specifically, $\text{Dep}(\boldsymbol{x}_t)$ is computed as the averaged mutual information across both feature-wise and component-wise slices of $\boldsymbol{x}_t$. A small value of $\text{Dep}(\boldsymbol{x}_t)$ indicates that the $\boldsymbol{x}_t$ is statistically independent. It thus provides a concrete condition under which GA becomes valid from the perspective of data dependency. The implementation details of $\text{Dep}(\boldsymbol{x}_t)$ are provided in Appendix F.1.

**Distributional Similarity.** While data dependency decay captures the disappearance of dependence, GA also requires that the marginals of $\boldsymbol{x}_t$ align with those of a Gaussian distribution. To assess this, we measure the distributional similarity between $\boldsymbol{x}_t$ and a reference Gaussian with matching variance using the Kolmogorov–Smirnov (KS) distance (Massey Jr, 1951). Concretely, for each dimension $\boldsymbol{x}_t^{(j)}$, we compare its empirical cumulative distribution function (CDF) $F_{t,j}(x)$ with the Gaussian CDF $\Phi_{\tilde{v}_t}(x)$, and average across all dimensions:

$$D_t = \frac{1}{d} \sum_{j=1}^{d} D_{t,j}, \quad \text{where} \quad D_{t,j} = \sup_x \left| F_{t,j}(x) - \Phi_{\tilde{v}_t}(x) \right|. \tag{12}$$

A smaller $D_t$ indicates closer alignment with Gaussian marginals and therefore stronger justification for approximation by $\mathcal{N}(\tilde{\mu}_t, \tilde{v}_t I)$. Since exact convergence only holds at the terminal prior $\boldsymbol{x}_1 \sim \mathcal{N}(\mathbf{0}, \mathbf{I})$, we adopt a tolerance $\varepsilon_{\text{DS}} > 0$ and define the distributional-similarity timestep as

$$T_{\text{DS}} := \min \left\{ t \,\Big|\, D_t \leq \varepsilon_{\text{DS}} \right\}. \tag{13}$$

**Estimation of $T^*$.** As established in Proposition 3.1, molecular data inherently distributes closer to Gaussian. From the joint perspectives of dependency decay and distributional similarity, we provide concrete and quantitative characterizations of the Gaussianity of $\boldsymbol{x}_t$, ensuring that the approximated $\boldsymbol{x}_{T^*}$ is sufficiently independent and marginally Gaussian. Accordingly, we define the operational GA timestep as

$$T^* = \max(T_{\text{ID}}, T_{\text{DS}}), \tag{14}$$

which guarantees that $\boldsymbol{x}_{T^*}$ satisfies both independence and marginal Gaussianity. The pseudo code of estimation algorithm is shown in Appendix E.

**Merits of GAGA.** Overall, GAGA provides three notable merits: **(1)** It improves computational efficiency of GPPGMs in both training and sampling by truncating the trajectory after the GA timestep $T^*$, thereby avoiding ineffective optimization and redundant inference on over-noised data. **(2)** It enhances generation quality owing to the removal of redundant trajectory segments without meaningful data information guidance, i.e., they have obtained sufficient Gaussianity. This strategy allows the model to concentrate on reconstructing the probability path in regions where structural information is still preserved. **(3)** GAGA is compatible with existing SDE/ODE solver-based accelerators such as DDIM (Song et al., 2020), and also improves generation quality when combined with them, as empirically validated in Sec. 4.4.

## 4 EXPERIMENTS

In this section, we empirically evaluate the proposed method on standard molecular generation benchmarks. We present the experimental setup, define the evaluation metrics, and report quantitative results on both generation quality and efficiency. Additional details on the Gaussianity tests and experimental configurations are provided in Appendix F and Appendix G, respectively.

### 4.1 EXPERIMENTAL SETUP

**Datasets.** We conduct experiments on widely-used molecular datasets, QM9 (Ramakrishnan et al., 2014) and GEOM-Drugs (Axelrod & Gomez-Bombarelli, 2022). QM9 contains 130K small molecules with up to 29 atoms, while GEOM-Drugs comprises 450K drug-like molecules with an average of 44 and up to 181 atoms. The configuration of datasets follows Hoogeboom et al. (2022) for regular generation and Xu et al. (2023) for latent-space generation, respectively.

**Baselines.** We conduct extensive experiments to compare with several state-of-the-art baselines. For a fair comparison, we restrict our baselines to those that do not explicitly incorporate bond information, either in data modeling (e.g., Irwin et al. (2025)) or in post-hoc processing (e.g., O'Boyle et al. (2011)). Specifically, G-SchNet (Gebauer et al., 2019) and Equivariant Normalizing Flows (ENF) (Garcia Satorras et al., 2021) employ autoregressive models for molecule generation. Equivariant Diffusion Model (EDM) (Hoogeboom et al., 2022), Geometric Latent Diffusion Model (GeoLDM) (Xu et al., 2023), and Equivariant Flow Matching model (EquiFM) (Song et al., 2023) are three representative GPPGMs from different perspectives for molecular generation, including regular diffusion, latent diffusion, and flow-matching. Moreover, the variants of EDM (GDM) and GeoLDM (GraphLDM) are also employed for comparison.

**Metrics.** We evaluate our method on standard molecular generation benchmarks using two broad classes of metrics: generation quality and efficiency. For generation quality, we report **validity** (the proportion of chemically valid molecules according to standard valency checks), **uniqueness** (the proportion of distinct molecules among generated samples), **molecular stability** (the fraction of generated molecules satisfying correct valency constraints), and **atom stability** (the fraction of generated atoms satisfying correct valency constraints). Following prior works (Xu et al., 2023; Hong et al., 2025), these metrics are computed via the cheminformatic tool RDKit (Landrum et al., 2016) over 10,000 generated samples. For efficiency, we record the average **sampling time (S-Time)** in GPU seconds per sample and total **training time (T-Time)** in GPU days. Both of them are measured on identical hardware across different baselines. Moreover, the **trajectory length (Steps)** $T^*$ is also shown in the results. These metrics collectively quantify the fidelity, diversity, and practical computational benefits of our method.

### 4.2 QUANTITATIVE PERFORMANCE

Table 1: Quantitative results on the QM9 dataset. The results of all baselines are directly obtained from their original works. The results of backbones applied with GAGA are highlighted with grey background. We run the evaluation 3 times and report the mean and standard deviation (std). Compared with previous methods, GA benefits all methods, achieving up to a 3.6% improvement in the molecule stability metric, and significantly reducing the generative trajectory length by 40%.

| Model | Generation Performance | | | | Efficiency | | |
| | Atom Sta (%) | Mol Sta (%) | Valid (%) | Valid * Uniq (%) | S-Time (GPU sec.) | T-Time (GPU day) | Traj. Len. (Steps) |
|---|---|---|---|---|---|---|---|
| Data | 99.0 | 95.2 | 97.7 | 97.7 | - | - | - |
| ENF | 85.0 | 4.9 | 40.2 | 39.4 | - | - | - |
| G-SchNet | 95.7 | 68.1 | 85.5 | 80.3 | - | - | - |
| GDM-AUG | 97.6 | 71.6 | 90.4 | 89.5 | 0.52 | 2.9 | 1000 |
| GraphLDM | 97.2 | 70.5 | 83.6 | 82.7 | 0.36 | 5.7 | 1000 |
| EDM | 98.7 | 82.0 | 91.9 | 90.7 | 0.65 | 5.6 | 1000 |
| EDM + GAGA | $98.9 \pm 0.03$ | $85.6 \pm 0.22$ | $94.7 \pm 0.04$ | $92.0 \pm 0.12$ | 0.36 | 3.1 | 550 |
| GeoLDM | 98.9 | 89.4 | 93.8 | 92.7 | 0.64 | 11.7 | 1000 |
| GeoLDM + GAGA | $99.2 \pm 0.01$ | $92.3 \pm 0.06$ | $96.7 \pm 0.14$ | $94.4 \pm 0.21$ | 0.42 | 7.2 | 650 |
| EquiFM | 98.9 | 88.3 | 94.7 | 93.5 | 0.17 | 6.2 | 200 |
| EquiFM + GAGA | $99.0 \pm 0.02$ | $91.2 \pm 0.11$ | $96.2 \pm 0.09$ | $93.7 \pm 0.18$ | 0.15 | 4.9 | 160 |

'-' denotes the invalid or not recorded setting in the original publication.

We evaluate the effectiveness of GAGA across multiple molecular generative baselines on both the QM9 and GEOM-Drugs datasets. The results on QM9 are reported in Table 1. We can observe from

the table that GAGA reduces both training and sampling costs by shortening the diffusion trajectory by up to 40%, and consistently improves generation quality. GAGA does not alter the granularity of the original generative path; rather, it leverages the early timestep of molecular data for obtaining sufficient Gaussianity, which enables the safe trajectory truncation. Therefore, the GPPGM could begin training and sampling from an earlier noise step without violating the probability path of the preserved segments. It also enables the model to focus on learning reconstructing trajectories where data information is still preserved, thereby improving the final generation quality.

Table 2: Quantitative results on GEOM dataset. We run the evaluation 3 times and report the mean and std. In general, GA improves generation performance and provides better efficiency across models. The results of backbones applied with GAGA are highlighted with grey background.

| Model | Generation Performance | | Efficiency | |
| | Atom Sta (%) | Valid (%) | S-Time (GPU sec.) | Traj. Len. (Steps) |
|---|---|---|---|---|
| Data | 86.5 | 99.9 | – | – |
| GDM-AUG | 77.7 | 91.8 | – | 1000 |
| GraphLDM | 76.2 | 97.2 | – | 1000 |
| EDM | 81.3 | 92.6 | 10.9 | 1000 |
| EDM + GAGA | $84.3 \pm 0.27$ | $93.4 \pm 0.49$ | 6.4 | 650 |
| GeoLDM | 84.4 | 99.3 | 10.2 | 1000 |
| GeoLDM + GAGA | $85.9 \pm 0.08$ | $99.3 \pm 0.03$ | 7.9 | 800 |

'–' denotes the invalid or not recorded setting in the original publication.

Similar benefits are observed on the more challenging GEOM-Drugs dataset. In this experiment, following prior work (Hoogeboom et al., 2022; Xu et al., 2023; Lipman et al., 2022), we omit metrics such as uniqueness, as it consistently remains close to 100% across all baselines. Similarly, we exclude molecule stability, which stays near 0% for all baselines. In general, GAGA consistently improves generation quality and reduces per-sample generation time across various baselines, as shown in Table 2. **These improvements on both generation quality and efficiency are particularly noteworthy given that GAGA requires no changes to model architecture. Instead, it modifies only the generative trajectory by leveraging the early timesteps for obtaining sufficient Gaussianity of molecular data.**

### 4.3 ABLATION STUDY

Table 3 presents an ablation study on the choice of GA timestep $T^*$. We evaluate models with both shorter and longer truncation points to investigate the impact of different $T^*$. We observe that applying GA too early ($T^* = 450$ for EDM and $T^* = 550$ for GeoLDM) results in efficiency gains but sacrifices the generation quality, since premature truncation collapses residual trajectories maintaining chemical plausibility and structural diversity, which are still beneficial for reconstruction of probability path. On the other hand, setting $T^*$ too late ($T^* = 650$ for EDM and $T^* = 750$ for GeoLDM) hurts generation quality while incurring unnec-

Table 3: Ablation study on the GA timestep $T^*$. We evaluate EDM and GeoLDM models under longer/shorter trajectories trained on QM9 dataset. We run the evaluation 3 times and report the mean value. The results with estimating $T^*$ via our proposed method are shown with grey background.

| Model | Valid * Uniq (%) | S-Time (GPU sec.) |
|---|---|---|
| EDM ($T^* = 1000$) | 90.7 | 0.65 |
| EDM + GAGA ($T^* = 450$) | 91.4 | 0.32 |
| EDM + GAGA ($T^* = 650$) | 91.6 | 0.45 |
| EDM + GAGA ($T^* = 550$) | 92.0 | 0.36 |
| GeoLDM ($T^* = 1000$) | 91.9 | 0.64 |
| GeoLDM + GAGA ($T^* = 550$) | 93.3 | 0.36 |
| GeoLDM + GAGA ($T^* = 750$) | 93.5 | 0.49 |
| GeoLDM + GAGA ($T^* = 650$) | 94.4 | 0.42 |

essary computational cost, since the data has already attained sufficient Gaussianity and additional steps mainly add redundant noise. **These results suggest that the optimal $T^*$ lies at the earliest point where Gaussianity is achieved, and truncation there preserves generation quality while offering the most substantial efficiency improvement.**

### 4.4 COMPATIBILITY WITH OTHER ACCELERATORS

Most existing acceleration algorithms for GPPGMs improve sampling efficiency by designing an efficient SDE/ODE solver applied in the backward process. Notable examples include DDIM (Song et al., 2020), DPM-Solver Series (Lu et al., 2022; Zheng et al., 2023; Lu et al., 2025), Gaussian-mixture solvers (Guo et al., 2023), etc. Since DDIM is the pioneering

and most widely adopted member of this family, we take it as a representative to study the compatibility of GAGA with optimized differential equation (DE) solver-based accelerators. Table 4 reports results for various backbones under DDIM with $2\times$ acceleration and in combination with GAGA. In addition to the generation improvement on backbone models, the experimental results also demonstrate

Table 4: Compatibility study of GAGA with DDIM ($2\times$ acceleration). We evaluate them 3 times and report the mean value on QM9 dataset. GAGA-applied backbone models are highlighted with grey background.

| Backbone | DDIM | GAGA | Valid * Uniq (%) | Traj. Len. (Steps) |
|---|---|---|---|---|
| EDM | ✗ | ✗ | 90.7 | 1000 |
| | ✗ | ✓ | 92.0 | 550 |
| | ✓ | ✗ | 83.7 | 500 |
| | ✓ | ✓ | 83.9 | 275 |
| GeoLDM | ✗ | ✗ | 91.9 | 1000 |
| | ✗ | ✓ | 94.4 | 650 |
| | ✓ | ✗ | 85.8 | 500 |
| | ✓ | ✓ | 87.5 | 325 |

that our GAGA is fully compatible with DE solvers. Compared with relying solely on DDIM, the joint configuration of DDIM and GAGA yields improved generation performance while inheriting efficiency benefits from both. Moreover, thanks to the preserved granularity of the generative trajectory, GAGA reduces the number of timesteps to a level comparable with DDIM, yet substantially outperforms it, as the coarsened trajectories induced by DDIM deteriorate generation quality[4]. **These results confirm that GAGA is orthogonal to other DE solver–based accelerators and can be seamlessly combined with them, while further enhancing generation quality. Importantly, GAGA also improves training efficiency, which is generally unattainable for most DE solver–based accelerators.**

## 5 RELATED WORK

**Probability Path based Generative Models (PPGMs).** PPGMs generate samples over data distributions by learning a transport process that maps simple prior distributions to complex data distributions through a sequence of structured transformations, i.e., the probability path. Specifically, diffusion-based generative models simulate this sequential transformation via SDEs, which have emerged as a powerful paradigm for multi-modal data synthesis (Croitoru et al., 2023; Kementzidis et al., 2025; Xu et al.). However, their iterative sampling (often requiring hundreds of steps) poses a significant speed bottleneck. A variety of techniques aim to accelerate diffusion sampling, such as progressive distillation (Salimans & Ho, 2022) and learned noising schedules (Williams et al., 2024). Nevertheless, the training process still typically requires hundreds of steps. Beyond diffusion models, Flow Matching offers a fresh perspective on acceleration. Flow Matching models train a continuous normalizing flow by regressing an optimal vector field along prescribed paths. Lipman et al. (2022) showed that using diffusion-style Gaussian paths in flow matching yields more robust training and faster ODE-based sampling. However, the nonlinear and high-curvature nature of learned transport fields makes it challenging to accurately approximate such trajectories with few discretization steps during training (Hassan et al., 2024; Eijkelboom et al., 2024).

**Gaussian Approximation (GA).** GA has long been a cornerstone in machine learning theory and practice. The Central Limit Theorem provides a classical justification: aggregates of many random factors tend toward a Gaussian distribution, which often explains why high-dimensional features or latent codes appear approximately normal (Hazra et al., 2021; Düker et al., 2024). Some researchers have explored the potential of the Gaussian approximation in generative modeling. For instance, Wang & Vastola observe that at high noise levels, the learned diffusion score can be well-approximated by a linear Gaussian model. Therefore, they can skip 15–30% of the sampling steps without degrading output fidelity. Such findings reinforce the idea that Gaussian assumptions can serve as an effective proxy for complex distributions in certain regimes, providing practical speedups without significant fidelity loss.

**Relation with Prior Works.** Improving the quality–efficiency tradeoff in generative molecular modeling has been approached from several directions. One line of work focuses on designing better

---

[4]Notably, we found higher-order accelerators, such as Lu et al. (2022; 2025), yield much worse performance than DDIM on molecular generation, because their assumption of smooth higher-order continuity makes them push the noised states toward similar trajectories, which significantly reduces molecular uniqueness.

forward process and understanding SNR behavior, including cosine schedules (Nichol & Dhariwal, 2021), Fourier-space schedule (Falck et al., 2025), ELBO-based augmentation (Kingma & Gao, 2023), and the unified perspective under ODE modeling (Gao et al., 2025). Another line proposes new generative trajectories tailored to molecular geometry, such as equivariant ODE flows (Song et al., 2023), Bayesian flow networks (Song et al., 2024), and straight-line diffusion (Ni et al., 2025). While these methods modify the forward or backward probability path to accelerate sampling or improve fidelity, our GAGA addresses the same goal from a different angle: we keep the original schedule and generative dynamics unchanged, and instead identify when the noised distribution has already reached the Gaussian regime so that redundant high-noise segments can be safely truncated. This design makes our method compatible and complementary to existing approaches. For instance, EDM (Hoogeboom et al., 2022) already adopt optimized noise schedule in forward process and EquiFM (Song et al., 2023) successfully reduced the trajectory to much fewer steps. However, our GAGA still improves generation quality while further obtaining better efficiency, as empirically verified in Sec. 4.2.

# 6 CONCLUSION AND FUTURE WORK

In this work, we introduced a principled framework, GAGA, for efficient GPPGMs on 3D molecular generation. By leveraging zero-mean preprocessing and empirical variance estimation, we proposed an analytic Gaussian approximation that identifies a characteristic time step $T^*$ at which the noised data obtained sufficient Gaussianity. This approximation enables the truncation of redundant generative trajectories, which are inefficient transport between "Gaussian-like" distributions. As a result, our GAGA can improve the efficiency of both sampling and training with consistent improvements in molecular generation quality.

**Limitations and Future Works.** Despite its empirical success, current GAGA assumes that the data modality is zero-mean invariant. While this assumption holds in many geometric and categorical domains, it is not valid for modalities like natural images or videos, where the absolute mean carries semantic information. Extending our methodology to such domains requires new techniques for determining Gaussianity without relying on zero-mean centering. In the future, we desire to design a unified framework of GAGA, which is capable of accommodating both zero-mean invariant and non-invariant modalities. It would significantly enhance the generalization capability of GAGA in probabilistic generative modeling. In addition, our Gaussianity evaluator may also serve as a tool for reallocating timesteps in quality-focused settings, offering a complementary research direction beyond acceleration.

## Ethics Statement

This work introduces a methodological contribution for improving the efficiency of 3D molecular generation models. All experiments are conducted on standard public benchmarks (QM9 and GEOM-Drugs) without sensitive or personally identifiable data. While molecular generative models have potential applications in drug discovery and material design, such use requires careful domain-specific safety and ethical oversight. Our contribution is limited to algorithmic efficiency and evaluation on benchmark datasets, and does not involve deployment to real-world systems.

## Reproducibility

We provide full details of our method, including theoretical derivations, Gaussianity evaluation procedures, and implementation settings, in the main text and appendices. All experiments are conducted on publicly available datasets (QM9 and GEOM-Drugs), and we report the exact model architectures, training configurations, and evaluation metrics. The code for algorithm computation is open-sourced. Together, these descriptions ensure that our results can be independently verified and extended by the community.

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

# Appendix

## A  USAGE OF LARGE LANGUAGE MODELS (LLMS)

In accordance with the ICLR 2026 policy on the use of LLMs, we disclose that large language models were employed solely as general-purpose writing assistants. Specifically, LLMs were only used to improve grammar, clarity, and style of exposition. All of the content assisted with LLMs is finally verified by authors.

## B  EXAMPLES AND ANALYSIS OF ZERO-MEAN INVARIANT DATA

We aim to show that for data modalities satisfying zero-mean invariance, the operation of zero-mean preserves all structural information relevant to generative modeling. We mainly discuss the Euclidean and non-uniform one-hot cases, which are tested in our experiments.

**Euclidean Data.** Let $\{\boldsymbol{x}_i\}_{i=1}^n \subset \mathbb{R}^d$ denote a collection of $n$ vectors (e.g., 3D Euclidean coordinates of atoms). Define the sample mean $\bar{\boldsymbol{x}} = \frac{1}{n}\sum_{i=1}^n \boldsymbol{x}_i$, and let $\tilde{\boldsymbol{x}}_i = \boldsymbol{x}_i - \bar{\boldsymbol{x}}$ be the centered representation. We claim that related positions are invariant under mean-centering:

$$\tilde{\boldsymbol{x}}_i - \tilde{\boldsymbol{x}}_j = (\boldsymbol{x}_i - \bar{\boldsymbol{x}}) - (\boldsymbol{x}_j - \bar{\boldsymbol{x}}) = \boldsymbol{x}_i - \boldsymbol{x}_j. \tag{15}$$

Hence, all geometric properties that depend on related positions, such as adjacency structures, bond lengths, or conformational shapes, are preserved exactly under centering. Consequently, zero-mean projection retains full information about the relational structure of the data.

**Non-Uniform One-Hot Categorical Vectors.** Let $h_i \in \{0,1\}^d$ denote a one-hot encoded vector satisfying $\sum_{j=1}^d (h_i)_j = 1$, and let $\bar{h} = \frac{1}{n}\sum_{i=1}^n h_i$ be the sample mean across a batch of $n$ such vectors. Define the centered vector $\tilde{h}_i = h_i - \bar{h}$. Note that each $\tilde{h}_i \in \mathbb{R}^d$ lies in a subspace orthogonal to the constant vector $\mathbf{1}_d$, since:

$$\sum_{j=1}^d (\tilde{h}_i)_j = \sum_{j=1}^d (h_i - \bar{h})_j = 1 - \sum_{j=1}^d \bar{h}_j = 0. \tag{16}$$

Moreover, the inner product between two centered vectors $\tilde{h}_i$ and $\tilde{h}_j$ satisfies:

$$\langle \tilde{h}_i, \tilde{h}_j \rangle = \langle h_i, h_j \rangle - \langle h_i, \bar{h} \rangle - \langle \bar{h}, h_j \rangle + \langle \bar{h}, \bar{h} \rangle, \tag{17}$$

from which it follows that pairwise centered dot products retain sufficient information to distinguish between original categorical identities once the category set is not degenerate (e.g., uniform). Since each one-hot vector $h_i$ is uniquely defined by a single active index, subtracting the global mean $\bar{h}$ merely induces a translation within the categorical simplex. The position of the maximal entry in $\tilde{h}_i$ still identifies the active class as long as $\bar{h}$ does not collapse distinct $h_i$ vectors onto the same centered value. Therefore, for any non-uniform categorical data embedded via one-hot encoding, mean-centering preserves the identity of the active component up to an affine transformation of the ambient space. As a result, zero-mean preprocessing retains the categorical semantics necessary for generative modeling under Euclidean approximation schemes.

## C  PROOF OF PROPOSITION 3.1

**Cumulants.** Let $X \in \mathbb{R}^d$ have moment generating function (mgf) $M_X(u) = \mathbb{E}[e^{u^\top X}]$ and cumulant generating function $K_X(u) = \log M_X(u)$, $u \in \mathbb{R}^d$. The $k$-*th cumulant tensor* is

$$\left( C^{(k)}(X) \right)_{i_1,\ldots,i_k} = \frac{\partial^k K_X(u)}{\partial u_{i_1} \cdots \partial u_{i_k}}\bigg|_{u=0}, \qquad k \geq 1.$$

In particular $C^{(1)}(X) = \mu := \mathbb{E}[X]$, $C^{(2)}(X) = \Sigma := \mathrm{Cov}(X)$, and $C^{(k)}(G) = 0$ for all $k \geq 3$ if $G$ is Gaussian.

**Setup.** Let $\{\boldsymbol{x}_t\}_{t=0}^T$ be the forward (noising) trajectory under a variance-preserving schedule. Fix $t \in [0, T)$ and $s > t$. Then

$$\boldsymbol{x}_s = \sqrt{\bar{\alpha}_{s|t}}\,\boldsymbol{x}_t + \sqrt{1 - \bar{\alpha}_{s|t}}\,\boldsymbol{\varepsilon}, \qquad \boldsymbol{\varepsilon} \sim \mathcal{N}(\mathbf{0}, \mathbf{I}),\ \ \boldsymbol{\varepsilon} \perp \boldsymbol{x}_t,\ \ \bar{\alpha}_{s|t} := \bar{\alpha}_s / \bar{\alpha}_t \in (0, 1). \tag{18}$$

**Gaussianity–and–independence functional (definition).** Let $\mathsf{D} := \{\mathrm{Diag}(v) : v \in \mathbb{R}^d\}$ be the diagonal subspace and define the orthogonal projections

$$\Pi_{\mathsf{D}}(\Sigma) := \mathrm{Diag}(\mathrm{diag}\,\Sigma), \qquad \Pi_{\mathsf{D}^\perp}(\Sigma) := \Sigma - \Pi_{\mathsf{D}}(\Sigma).$$

For weights $\beta > 0$ and $w_k > 0$ ($k \geq 3$), define for any random vector $x \in \mathbb{R}^d$

$$\mathcal{H}^{(K)}(x) := \beta\big\|\Pi_{D^\perp}(\mathrm{Cov}(x))\big\|_F + \mathbf{1}_{\{K \geq 3\}} \sum_{k=3}^K w_k \big\|C^{(k)}(x)\big\|_F. \tag{19}$$

In following analysis, we will assume $k \geq 3$, then we can omit the indicator function $\mathbf{1}_{\{\cdot\}}$. And the conclusion can be easily applied to $k = 2$.

**Lemma C.1** (Variance preserving propagation of moments/cumulants and contraction of $\mathcal{H}^{(K)}$). *Let $t \in [0, T)$ and $s > t$, and write $a := \bar{\alpha}_{s|t} \in (0, 1)$. Under equation 18, with $\Sigma_t := \mathrm{Cov}(\boldsymbol{x}_t)$ and $B_{k,t} := \big\|C^{(k)}(\boldsymbol{x}_t)\big\|_F$ for $k \geq 3$,*

$$\mu_s = \sqrt{a}\,\mu_t, \qquad \Sigma_s = a\,\Sigma_t + (1 - a)\,\mathbf{I}, \qquad \big\|C^{(k)}(\boldsymbol{x}_s)\big\|_F = a^{k/2} B_{k,t}\ \ (k \geq 3).$$

*Consequently,*

$$\mathcal{H}^{(K)}(\boldsymbol{x}_s) = \beta\,a\,\big\|\Pi_{\mathsf{D}^\perp}(\Sigma_t)\big\|_F + \sum_{k=3}^K w_k\,a^{k/2}\,B_{k,t}, \tag{20}$$

*and*

$$\frac{\partial}{\partial a}\,\mathcal{H}^{(K)}(\boldsymbol{x}_s) = \beta\,\big\|\Pi_{\mathsf{D}^\perp}(\Sigma_t)\big\|_F + \sum_{k=3}^K w_k\,\frac{k}{2}\,a^{k/2-1}\,B_{k,t}\ >\ 0$$

*whenever $\big\|\Pi_{\mathsf{D}^\perp}(\Sigma_t)\big\|_F + \sum_{k=3}^K B_{k,t} > 0$. Hence, since $a = \bar{\alpha}_{s|t}$ decreases strictly in $s$ for a VP schedule, $\mathcal{H}^{(K)}(\boldsymbol{x}_s)$ is strictly decreasing in $s$ unless $\boldsymbol{x}_t$ is already an independent Gaussian (in which case $\mathcal{H}^{(K)}(\boldsymbol{x}_s) \equiv 0$).*

*Proof.* First, $\mu_s = \mathbb{E}[\boldsymbol{x}_s] = \sqrt{a}\,\mu_t + \sqrt{1-a}\,\mathbb{E}[\boldsymbol{\varepsilon}] = \sqrt{a}\,\mu_t$. For the covariance, write $\boldsymbol{x}_s = \sqrt{a}\,\boldsymbol{x}_t + \sqrt{1-a}\,\boldsymbol{\varepsilon}$ and center by the means:

$$\boldsymbol{x}_s - \mu_s = \sqrt{a}\,(\boldsymbol{x}_t - \mu_t) + \sqrt{1-a}\,\boldsymbol{\varepsilon}.$$

Independence and $\mathbb{E}[\boldsymbol{\varepsilon}] = 0$ give

$$\Sigma_s = \mathbb{E}\big[(\boldsymbol{x}_s - \mu_s)(\boldsymbol{x}_s - \mu_s)^\top\big] = a\,\Sigma_t + (1 - a)\,\mathbb{E}[\boldsymbol{\varepsilon}\boldsymbol{\varepsilon}^\top] = a\,\Sigma_t + (1 - a)\,\mathbf{I}.$$

Linearity of $\Pi_{\mathsf{D}^\perp}$ and $\mathbf{I} \in \mathsf{D}$ yield

$$\Pi_{\mathsf{D}^\perp}(\Sigma_s) = \Pi_{\mathsf{D}^\perp}(a\,\Sigma_t) + (1 - a)\,\Pi_{\mathsf{D}^\perp}(\mathbf{I}) = a\,\Pi_{\mathsf{D}^\perp}(\Sigma_t),$$

hence $\|\Pi_{\mathsf{D}^\perp}(\Sigma_s)\|_F = a\,\|\Pi_{\mathsf{D}^\perp}(\Sigma_t)\|_F$.

For cumulants, independence implies additivity: $C^{(k)}(X + Y) = C^{(k)}(X) + C^{(k)}(Y)$ when $X \perp Y$ (this follows from $K_{X+Y}(u) = K_X(u) + K_Y(u)$). Homogeneity follows from $K_{cX}(u) = \log \mathbb{E}[e^{u^\top cX}] = K_X(cu)$ and the chain rule:

$$\frac{\partial^k}{\partial u_{i_1} \cdots \partial u_{i_k}} K_{cX}(u)\bigg|_{u=0} = c^k\,\frac{\partial^k}{\partial u_{i_1} \cdots \partial u_{i_k}} K_X(u)\bigg|_{u=0} \ \Rightarrow\ C^{(k)}(cX) = c^k C^{(k)}(X).$$

Because a Gaussian has $C^{(k)}(\boldsymbol{\varepsilon}) = 0$ for $k \geq 3$,

$$C^{(k)}(\boldsymbol{x}_s) = C^{(k)}(\sqrt{a}\,\boldsymbol{x}_t) + C^{(k)}(\sqrt{1-a}\,\boldsymbol{\varepsilon}) = a^{k/2} C^{(k)}(\boldsymbol{x}_t),$$

so $\|C^{(k)}(\boldsymbol{x}_s)\|_F = a^{k/2}\|C^{(k)}(\boldsymbol{x}_t)\|_F$. Plugging these identities into equation 19 gives equation 20. Finally, since $\beta > 0, w_k > 0$ and $B_{k,t} \geq 0$, the displayed derivative is $> 0$ whenever not all terms vanish. As $a$ strictly decreases in $s$ for VP, $\mathcal{H}^{(K)}(\boldsymbol{x}_s)$ strictly decreases in $s$ unless already identically zero. $\quad\square$

**Lemma C.2** ($\theta$-decomposition via prefix sums). *For $a \in (0,1)$, define*

$$\theta_2(a) := a - a^{3/2}, \qquad \theta_m(a) := a^{m/2} - a^{(m+1)/2} \ (3 \le m \le K-1), \qquad \theta_K(a) := a^{K/2},$$

*and for $m \in \{2,3,\ldots,K\}$ define the* prefix functionals

$$\mathcal{H}^{(m)}(\boldsymbol{x}_t) := \beta \left\| \Pi_{\mathrm{D}^\perp}(\Sigma_t) \right\|_F + \sum_{k=3}^{m} w_k \left\| C^{(k)}(\boldsymbol{x}_t) \right\|_F.$$

*Then $\theta_m(a) > 0$ for all $m$ and $a \in (0,1)$, and the closed form equation 20 admits*

$$\mathcal{H}^{(K)}(\boldsymbol{x}_s) = \sum_{m=2}^{K} \theta_m(a) \, \mathcal{H}^{(m)}(\boldsymbol{x}_t), \qquad \text{with} \quad \sum_{m=j}^{K} \theta_m(a) = a^{j/2} \text{ for each } j \in \{2,3,\ldots,K\}. \tag{21}$$

*Proof.* For $0 < a < 1$, $\theta_m(a) = a^{m/2}(1 - a^{1/2}) > 0$ for $m \le K-1$ and $\theta_K(a) = a^{K/2} > 0$. To prove equation 21, expand the right-hand side:

$$\sum_{m=2}^{K} \theta_m(a) \, \mathcal{H}^{(m)}(\boldsymbol{x}_t) = \Big( \sum_{m=2}^{K} \theta_m(a) \Big) \beta \|\Pi_{\mathrm{D}^\perp}(\Sigma_t)\|_F + \sum_{k=3}^{K} \Big( \sum_{m=k}^{K} \theta_m(a) \Big) w_k \|C^{(k)}(\boldsymbol{x}_t)\|_F.$$

Hence it suffices to show the *tail-sum identities* $\sum_{m=2}^{K} \theta_m(a) = a$ and $\sum_{m=k}^{K} \theta_m(a) = a^{k/2}$ for each $k \in \{3,\ldots,K\}$. For $k \le K-1$,

$$\sum_{m=k}^{K-1} \big(a^{m/2} - a^{(m+1)/2}\big) + a^{K/2} = \big(a^{k/2} - a^{(k+1)/2}\big) + \cdots + \big(a^{(K-1)/2} - a^{K/2}\big) + a^{K/2} = a^{k/2},$$

a telescoping sum; the case $k = K$ is immediate. The identity for $k = 2$ is the same computation with $k = 2$. Substituting these tail-sums into the expansion recovers equation 20. $\square$

**Lemma C.3** (Order preservation under prefix dominance). *Let $A, B$ be two classes of data set at time $t$. Assume the* prefix dominance

$$\mathcal{H}^{(m)}(\boldsymbol{x}_t^A) \le \mathcal{H}^{(m)}(\boldsymbol{x}_t^B) \quad \text{for all } m = 2,3,\ldots,K, \tag{22}$$

*with at least one strict inequality. Then, for every $s > t$ (equivalently, every $a \in (0,1)$),*

$$\mathcal{H}^{(K)}(\boldsymbol{x}_s^A) = \sum_{m=2}^{K} \theta_m(a) \, \mathcal{H}^{(m)}(\boldsymbol{x}_t^A) < \sum_{m=2}^{K} \theta_m(a) \, \mathcal{H}^{(m)}(\boldsymbol{x}_t^B) = \mathcal{H}^{(K)}(\boldsymbol{x}_s^B),$$

*and the inequality is strict because all $\theta_m(a) > 0$ for $a \in (0,1)$.*

*Proof.* By Lemma C.2, $\mathcal{H}^{(K)}(\boldsymbol{x}_s) = \sum_{m=2}^{K} \theta_m(a) \, \mathcal{H}^{(m)}(\boldsymbol{x}_t)$ with $\theta_m(a) > 0$. Applying equation 22 termwise gives $\mathcal{H}^{(K)}(\boldsymbol{x}_s^A) \le \mathcal{H}^{(K)}(\boldsymbol{x}_s^B)$. Strictness follows because at least one index $m^\star$ satisfies $\mathcal{H}^{(m^\star)}(\boldsymbol{x}_t^A) < \mathcal{H}^{(m^\star)}(\boldsymbol{x}_t^B)$ and $\theta_{m^\star}(a) > 0$, hence the weighted sum is strictly smaller. $\square$

Lemma C.1 shows that for each initialization, $s \mapsto \mathcal{H}^{(K)}(\boldsymbol{x}_s)$ is strictly decreasing (unless already at an independent Gaussian). Lemma C.3 states that if $A$ is *prefix-dominant* over $B$ at time $t$, then $\mathcal{H}^{(K)}(\boldsymbol{x}_s^A) < \mathcal{H}^{(K)}(\boldsymbol{x}_s^B)$ for every $s > t$. Therefore, for any threshold $\varepsilon > 0$, the hitting times

$$T_X(\varepsilon) := \inf\{s > t : \mathcal{H}^{(K)}(\boldsymbol{x}_s^X) \le \varepsilon\}$$

satisfy $T_A(\varepsilon) < T_B(\varepsilon)$, which formalizes that under VP the speed to gain sufficient Gaussianity for $A$ is faster than for $B$ whenever $A$ starts closer to Gaussian in the sense of equation 22.

Moreover, if there exist nondecreasing functions $\varphi_{\mathrm{dep}}, \varphi_{\mathrm{ks}} : [0,\infty) \to [0,\infty)$ with $\varphi_{\mathrm{dep}}(0) = \varphi_{\mathrm{ks}}(0) = 0$ such that for every $s > t$,

$$\mathrm{Dep}(\boldsymbol{x}_s) \le \varphi_{\mathrm{dep}}\big(\mathcal{H}^{(K)}(\boldsymbol{x}_s)\big), \qquad D(\boldsymbol{x}_s) \le \varphi_{\mathrm{ks}}\big(\mathcal{H}^{(K)}(\boldsymbol{x}_s)\big), \tag{23}$$

then, for any tolerances $\varepsilon_{\mathrm{dep}}, \varepsilon_{\mathrm{DS}} > 0$,

$$T_{\mathrm{ID}}^A := \inf\{s > t : \mathrm{Dep}(\boldsymbol{x}_s^A) \le \varepsilon_{\mathrm{dep}}\} \le T_{\mathrm{ID}}^B, \qquad T_{\mathrm{DS}}^A := \inf\{s > t : D(\boldsymbol{x}_s^A) \le \varepsilon_{\mathrm{DS}}\} \le T_{\mathrm{DS}}^B,$$

and hence $T_A^* := \max(T_{\mathrm{ID}}^A, T_{\mathrm{DS}}^A) \le T_B^*$, with strict inequality if equation 22 is strict for some $m$.

# D GAUSSIANITY COMPARISON BETWEEN IMAGE AND MOLECULAR DATA DISTRIBUTION

Table 5: Gaussianity comparison between QM9 molecular data and CIFAR-10 image data. All values are reported as mean $\pm$ variance across 10000 samples, which represent the first 10000 molecules of QM9 training dataset and 10000 images in CIFAR-10 test dataset.

| Dataset | KS Distance | MI (total) |
|---------|-------------|------------|
| QM9 | $2.77 \times 10^{-1} \pm 4.26 \times 10^{-4}$ | $1.66 \times 10^{-1} \pm 1.10 \times 10^{-3}$ |
| CIFAR-10 | $5.33 \times 10^{-1} \pm 1.04 \times 10^{-3}$ | $8.45 \times 10^{-1} \pm 1.84 \times 10^{-1}$ |

**Experimental Setup.** We compute two Gaussianity indicators on the raw data distribution of QM9 molecules and CIFAR-10 (Krizhevsky et al., 2009) images: (1) the Kolmogorov–Smirnov (KS) p-value, which measures the goodness-of-fit to a Gaussian reference ($\mathcal{N}(\mathbf{0}, \tilde{v}\,\mathbf{I})$ for QM9 and $\mathcal{N}(\mathbf{0}, \mathbf{I})$ for CIFAR-10); and (2) $\mathrm{MI_{total}} = \frac{\mathrm{MI_{feat}} + \mathrm{MI_{comp}}}{2}$, which captures the remaining statistical dependency structure that deviates from Gaussianity. Specifically, $\mathrm{MI_{feat}}$ represents the mutual information(MI) across features, e.g., the atomic positions and types of a molecule. The $\mathrm{MI_{comp}}$ is the MI across components, e.g., different atoms of a molecule. All datasets are first normalized before comparison, the molecular normalization follows (Hoogeboom et al., 2022), while the image normalization follows (Ho et al., 2020).

**Analysis.** As shown in Table 5, both distributional and dependency-based metrics indicate that molecular data are substantially closer to Gaussian at initialization than image data. In terms of marginal distributional similarity, QM9 exhibits a significantly smaller Kolmogorov–Smirnov (KS) distance to the Gaussian reference compared to CIFAR-10, implying that its empirical marginals deviate less from Gaussianity. From the perspective of dependency structure, the total mutual information (MI), defined as the average of feature-wise and sample-wise MI, is markedly lower for QM9, reflecting weaker statistical dependence among dimensions. In contrast, CIFAR-10 retains pronounced non-Gaussian characteristics, manifested by both a larger KS distance and substantially higher total MI. Together, these results confirm that molecular data distributions are inherently closer to Gaussian than natural image data, which explains why molecular trajectories attain sufficient Gaussianity at earlier noising steps.

# E PSEUDO CODE OF ESTIMATION $T^*$ IN GAGA

---

**Algorithm 1:** Pseudo Code of Estimation $T^*$ in GAGA

---

**Input:** Forward noised trajectory $\{x_t\}_{t \in [0,1]}$, which is discretized to $T$ steps; data dependency threshold $\epsilon_{\mathrm{dep}}$; distributional similarity threshold $\epsilon_{\mathrm{DS}}$; statistical test frequency $\lambda \in (0, 1]$.

**Output:** Estimated GA timestep $T^*$.

**Initialize:** $T_{\mathrm{ID}} \leftarrow \infty$, $T_{\mathrm{DS}} \leftarrow \infty$, counter $\mathsf{cnt} \leftarrow 0$.

**for** $t = \lambda T, 2\lambda T, \ldots, T$ **do**
  Compute dependency score $\mathrm{Dep}(x_t)$.
  Compute KS distance $D_t$ between $x_t$ and $\mathcal{N}(0, \tilde{v}_t I)$.
  **if** $\mathrm{Dep}(x_t) \leq \epsilon_{\mathrm{dep}}$ **and** $D_t \leq \epsilon_{\mathrm{DS}}$ **then**
    Increase $\mathsf{cnt} \leftarrow \mathsf{cnt} + 1$.
    **if** $\mathsf{cnt} \geq 2$ **then**
      $\llcorner$ Record $T^* \leftarrow t$ and **break**.
  **else**
    $\llcorner$ Reset $\mathsf{cnt} \leftarrow 0$.
  // Two consecutive passes to ensure the statistical
     reliability.

**return** $T^*$.

---

# F  GAUSSIANITY TEST DETAILS

## F.1  DATA DEPENDENCY TEST

By treating $\boldsymbol{x}_t$ as a sample from an intractable noised data distribution, we estimate the empirical mutual information (MI) across both the sample-wise and feature-wise dimensions of the data tensor. MI quantifies the degree of statistical dependency between random variables by measuring the divergence between their joint distribution and the product of their marginals. For two random vectors $X$ and $Y$, the mutual information is defined as:

$$I(X;Y) = \int \int p_{X,Y}(x,y) \log \frac{p_{X,Y}(x,y)}{p_X(x)\,p_Y(y)} \, dx \, dy, \tag{24}$$

where $p_{X,Y}(x,y)$ denotes the joint probability density, and $p_X(x), p_Y(y)$ are the marginal densities of $X$ and $Y$, respectively.

Given a data matrix representation of $\boldsymbol{x}_t \in \mathbb{R}^{N \times d}$, we estimate the dependency from two complementary perspectives: *feature-wise* and *component-wise*. In molecular data, $N$ is the number of atoms, and $d$ includes both 3D coordinates and atom-type features. Feature-wise MI evaluates the dependency within each atom's feature vector (e.g., correlations among its coordinate and type entries), while component-wise MI evaluates the dependency across atoms for each feature dimension (e.g., correlations among all atoms' 3D Euclidean coordinates). Formally, let

$$\mathcal{X}_{\text{feat}} = \{x_i \in \mathbb{R}^d : i = 1,\ldots,N\}, \qquad \mathcal{X}_{\text{comp}} = \{x^{(j)} \in \mathbb{R}^N : j = 1,\ldots,d\}, \tag{25}$$

denote the sets of feature-wise and component-wise slices, respectively. Then the empirical MI scores are given by

$$\text{MI}_{\text{feat}} = \frac{1}{|\mathcal{X}_{\text{feat}}|} \sum_{x \in \mathcal{X}_{\text{feat}}} I(x), \qquad \text{MI}_{\text{comp}} = \frac{1}{|\mathcal{X}_{\text{comp}}|} \sum_{x \in \mathcal{X}_{\text{comp}}} I(x), \tag{26}$$

where $I(x)$ denotes the estimated mutual information of the given vector $x$ across its components. The data dependency is then estimated by $Dep(\boldsymbol{x}) = \frac{\text{MI}_{\text{feat}} + \text{MI}_{\text{comp}}}{2}$. As $t$ increases, these MI statistics decay toward zero, indicating diminishing dependency and the emergence of approximate independence in $\boldsymbol{x}_t$. In our experiments, we set the MI threshold $\varepsilon_{\text{dep}} = 0.1$.

## F.2  DISTRIBUTIONAL SIMILARITY VIA KS-TEST

To evaluate whether the noised data $x_t$ has become sufficiently similar to the Gaussian distribution $\mathcal{N}(0, \tilde{v}_t I)$, we perform statistical testing based on the Kolmogorov–Smirnov (KS) criterion. At each test timestep $t$, we apply the one-sample KS test dimension-wise to the components of $x_t$ after zero-centering, treating each variable as an independent sample drawn from the empirical distribution. Specifically, for each dimension $j \in \{1,\ldots,d\}$, we compute the empirical cumulative distribution function (CDF) $F_{t,j}(x)$ and compare it against the theoretical CDF $\Phi_{\tilde{v}_t}(x)$ of a univariate normal distribution with zero mean and variance $\tilde{v}_t$, derived analytically from the forward noise schedule. The test statistic is defined as:

$$D_{t,j} = \sup_x |F_{t,j}(x) - \Phi_{\tilde{v}_t}(x)| . \tag{27}$$

For each coordinate $j$, we test $H_0 : x_t^{(j)} \sim \mathcal{N}(0, \tilde{v}_t)$ against $H_1 : x_t^{(j)} \not\sim \mathcal{N}(0, \tilde{v}_t)$. Under $H_0$, with sample size $n$, the scaled statistic $\sqrt{n}\, D_{t,j}$ converges to the Kolmogorov distribution with CDF $1 - 2\sum_{k=1}^{\infty}(-1)^{k-1} \exp(-2k^2\lambda^2)$, yielding the 5% critical threshold $D_{t,j} > c_{0.05}/\sqrt{n}$ (with $c_{0.05} \approx 1.36$ asymptotically). We declare that timestep $t$ satisfies the Gaussianity criterion if at least 95% of coordinates fail to reject $H_0$, i.e., $x_t$ is statistically indistinguishable from the reference $\mathcal{N}(0, \tilde{v}_t I)$ at 95% confidence level.

**Computational Complexity and Model-Agnostic Property.** Both the data dependency and distributional similarity tests in our GAGA scale on average as $\mathcal{O}(N \log N)$ with respect to the number of samples $N$. Since these tests are performed only once prior to training, their cost is negligible compared to the iterative computation required for model training over hundreds of diffusion steps. Moreover, the evaluation is fully *model-agnostic*: On each tested timestep, the injected noise is sampled independently; once the Gaussianity timestep $T^*$ is identified, it can be fixed and reused across different generative backbones without any need for re-computation.

## G  EXPERIMENTAL SETTINGS

### G.1  BACKBONE MODEL

In our experiments, all molecular generation baselines utilize the Equivariant Graph Neural Network (EGNN) (Satorras et al., 2021) as the backbone architecture for generative processing. EGNN operates on graphs embedded in Euclidean space and are designed to be equivariant under rigid-body transformations from the special Euclidean group $SE(3)$, including rotations and translations. This property ensures that molecular outputs transform consistently with the input geometry, preserving critical physical symmetries.

Formally, consider a molecule represented as a fully connected graph with $N$ nodes, where each node $i$ has coordinates $\boldsymbol{x}_i \in \mathbb{R}^3$ and associated atom features $\boldsymbol{h}_i \in \mathbb{R}^d$. At each EGNN layer, node features and positions are updated through message-passing operations:

$$
\begin{aligned}
\boldsymbol{m}_{ij} &= \phi_e(\boldsymbol{h}_i, \boldsymbol{h}_j, \|\boldsymbol{x}_i - \boldsymbol{x}_j\|^2), \\
\boldsymbol{h}'_i &= \phi_h\left(\boldsymbol{h}_i, \sum_{j \neq i} \alpha_{ij} \boldsymbol{m}_{ij}\right), \\
\boldsymbol{x}'_i &= \boldsymbol{x}_i + \sum_{j \neq i} \frac{\boldsymbol{x}_i - \boldsymbol{x}_j}{\|\boldsymbol{x}_i - \boldsymbol{x}_j\| + \epsilon} \phi_x(\boldsymbol{h}_i, \boldsymbol{h}_j, \|\boldsymbol{x}_i - \boldsymbol{x}_j\|^2),
\end{aligned}
\tag{28}
$$

where $\phi_e$, $\phi_h$, and $\phi_x$ are learnable functions (typically MLPs), and $\alpha_{ij}$ is an optional attention or reweighting term. $\boldsymbol{m}_{ij}$ is the passed message. The update rule guarantees that output features are equivariant with respect to $SE(3)$ transformations. This equivariant structure is critical for molecular generative tasks, as the physical properties of molecules are invariant to coordinate shifts and rotations.

### G.2  IMPLEMENTATION DETAILS

For all baseline models, we follow the official open-sourced codebases and retain their default hyperparameters unless otherwise specified. Gaussian Approximation is applied after the truncation step $T^*$, as estimated via our KS and MI-based Gaussianity evaluation. Moreover, to investigate the effect of GAGA on training, we retrain EDM and GeoLDM using truncated generation trajectories, i.e., the forward noise steps are limited to $T^*$ while reducing the training epoch identical in proportion to truncations of trajectories. For instance, in the original EDM (Hoogeboom et al., 2022) configuration on the QM9 dataset, the model is trained with a forward trajectory of 1000 steps over 3000 epochs. After applying GAGA, the estimated Gaussian-approximation timestep is $T^* = 550$, which reduces the maximum noising level used during training. To maintain comparable training sufficiency under this truncated trajectory, we scale the training epochs proportionally to the length of the preserved trajectory, i.e., from 3000 epochs to $3000 \times \frac{550}{1000} = 1650$ epochs. Consequently, the sampling will also start from 550 steps with the analytically derived Gaussian. This adjustment ensures that the model receives an equivalent amount of optimization per effective timestep, while still benefiting from a substantially shorter training trajectory. Notably, even under this reduced training budget and without modifying any architectural or hyperparameter settings of the baseline models, GAGA consistently improves both generation quality and efficiency over the original versions.

All molecular generation evaluation metrics are computed on 10,000 generated molecules using RDKit (Landrum et al., 2016). Validity and atom stability are defined by valency correctness, and uniqueness is computed as the percentage of distinct canonical SMILES. Sampling time is measured as the average GPU seconds to generate one molecule, while training time reflects total GPU days until the last pre-defined epochs in the official repositories.

All experiments are conducted on a computing cluster equipped with NVIDIA RTX 3090 GPUs, each with 24 GB memory. Training is parallelized across 2 GPUs using PyTorch DDP framework, while inference experiments are executed on a single GPU for fair comparison of sampling speed. The CPUs are Intel(R) Core(TM) i9-12900KF. Unless otherwise specified, we report sampling time as the average GPU seconds per generated sample, and training time in GPU days until the max epochs from the baselines' official repositories. All baseline implementations use their official code, pre-trained weights (if available) and hyperparameters to ensure comparability.

