# OpenReview forum: "GAGA: Gaussianity-Aware Gaussian Approximation for Efficient 3D Molecular Generation"
_ICLR.cc/2026/Conference — ICLR 2026 Poster_

### Official Review · Reviewer_FhFk · 2025-10-30

**Soundness:** 3
**Presentation:** 2
**Contribution:** 2
**Rating:** 2
**Confidence:** 3

**Summary:**

This paper aims to improve the training and sampling efficiency of diffusion models by truncating the diffusion noise-adding process. For zero-mean data, the authors design an empirical method to select diffusion timesteps, which is validated on molecular generation tasks. Experimental results show performance improvements over methods like EDM and GeoLDM.

**Strengths:**

1. The proposed method is simple and serves as a "free lunch": it eliminates redundant training and sampling overhead in existing models, boosting both efficiency and generation quality.

2. The writing is clear and easy to follow

**Weaknesses:**

1. The core idea of truncating the noise-adding process is overly straightforward. Similar concepts have already been explored in iDDPM, where the redundancy of original diffusion schedules was identified and new schedules were designed. This method essentially truncates the existing schedule, which introduces two critical issues:
    - Theoretical inconsistency: Truncation increases the approximation error of the Gaussian prior. The distance between $x_{T^*}$ and the standard Gaussian is larger than that between original $x_T$ and the standard Gaussian.
    - Practical inconvenience: A new hyperparameter $T^*$ (or two hyperparameters $\epsilon_{dep}$, $\epsilon_{DS}$) is introduced , which needs to tune through experiments.

Instead of truncation, a more viable approach could be adjusting the SNR decay rate of the schedule to slow down decay, thereby compressing the time interval during which the distribution approximates a Gaussian.

2. Limited significance of the method: Performance improvements are more pronounced for models with DDPM schedule (e.g., EDM, GeoLDM). Intuitively, the DDPM schedule has a fast SNR decay rate, leading to many high-noise timesteps—hence the truncation method yields more obvious gains. However, for methods that already address schedule redundancy (e.g., GeoBFN, SLDM), the efficiency improvements from this approach are expected to be far more limited.

3. Gaps in research logic and writing:
    - The paper introduces three metrics to measure the discrepancy between the current distribution and a Gaussian distribution: cumulant tensors (Eq. 8), mutual information (Eq. 12), and cumulative distribution functions (Eq. 14). The theoretical analysis relies on the first metric, while the second and third are used in experiments—creating a logical gap. Is there a need to include so many similar metrics? Would retaining the least computationally expensive one suffice?
    - Eq. 26 lacks implementation details for \text{MI}_{\text{rows}}: Does it compute the mutual information between all pairs of rows and then average the results? Eq. 12: Does it use row-wise or column-wise mutual information? What is the physical meaning of rows and columns? For the matrix representation of $x_t$: If $x_t$ represents coordinates, is the matrix of size $N \times 3$ (where N is the number of atoms) or $(3N) \times d_{\text{feat}}$ (where $d_{\text{feat}}$ is the feature dimension)? If the latter, does the independence measured by mutual information depend on the network used to extract features?
    - Unclear visualization in Figure 3: The "Gaussianity" in Figure 3 is represented by colors; numerical values would be more concrete.

Ref:
Improved Denoising Diffusion Probabilistic Models.

Straight-Line Diffusion Model for Efficient 3D Molecular Generation.

Unified Generative Modeling of 3D Molecules via Bayesian Flow Networks.

**Questions:**

1. Is the zero-mean assumption a major limitation of your method? Could normalization be applied to handle data with non-zero means, regardless of the original data distribution?
2. What would be the performance if timestep truncation is applied only during sampling (not training)? Would this harm generation quality?
3. A curious question: for element types alone, is their distribution closer to a Gaussian than that of images?

---

> ### Author Response · Authors · 2025-11-19
> **Response to Reviewer FhFk - Part I**
>
> Thank you for your insightful comments and thoughtful suggestions. We have revised the manuscript accordingly. Below, we provide point-by-point responses to your feedback.
>
> ### Weaknesses
>
> - W1:  The core idea of truncating the noise-adding process is overly straightforward compared with iDDPM, Theoretical inconsistency of GA, and Practical inconvenience about hyperparameters.
>
>   - Relationship with adjusting the SNR decay rate: We agree that SNR formulations, such as those explored in iDDPM and follow-up works, provide important insights into diffusion trajectory design, and **we now discuss these connections in the Sec. 5.** However, our contribution is **orthogonal to SNR-based schedule modification**. Although the works like iDDPM done something similar to truncation, and then designs new SNR curves to reshape the forward trajectory, **our method keeps the schedule unchanged and instead identifies the point where the existing schedule has already driven the data into the Gaussian regime**. Notably, **EDM itself adopts a cosine-inspired schedule precisely** because the iDDPM schedule provides good Gaussianization properties. **Even under this optimized schedule, our method offers large improvements.** This suggests that, it is difficult to design a universally optimal forward schedule for arbitrary data distributions, and schedule design alone cannot fully eliminate redundant over-noising. **SNR-based methods sugget that adjusting SNR decay may help the forward process reach Gaussianity more smoothly. While our method then automatically detects where Gaussianity is actually reached and truncates only when justified. Thus, our method is compatible with and beneficial on top of existing schedule-design approaches.**
>
>   - Theoretical inconsistency: We agree that truncating the forward trajectory inevitably introduces approximation error, since $X_{T^*}$ is not as close to the Gaussian prior as $X_T$. However, **this limitation is inherent to any Gaussian approximation (GA) technique, including those widely used in variational inference and high-dimensional problems.** The key question is **whether the approximation error is harmful in practice.** Our statistical tests, along with the empirical results across EDM, GeoLDM, and EquiFM consistently show that the GA in our method lead to (i) reduced over-noised training segments, (ii) improved learning focus on identity-preserving regions, and (iii) higher sample quality. In other words, the approximation error forms a beneficial trade-off that boosts both efficiency and generative fidelity in practice. Importantly, our framework does not alter the preserved portion of the probability path. **We truncate only after the trajectory has already entered the Gaussian regime**, where the remaining steps primarily transport between distributions that are already close to Gaussian. For this region, we believe the analytic GA is theoretically justified and practically effective.
>
>   - Practical inconvenience: We acknowledge that out method introduces thresholds $\epsilon_{\mathrm{dep}}$ and $\epsilon_{\mathrm{DS}}$. However, in practice this does not add burden:
>    **1. The Gaussianity test is performed once per dataset and schedule, not per model.**
>    **2. For any two GPPGMs taking the same input with the same noise schedule, the same $T^*$ applies without re-estimation.**
>
>     Thus, our method does not introduce tunable model-specific hyperparameters, and in practice behaves like a **dataset-dependent** preprocessing step. We clarify this point in Appendix F.2.
>
>
> - W2: Improvements stem mainly from ''DDPM schedules'' and connection with models that explicitly address schedule redundancy (GeoBFN, SLDM, EquiFM, etc.)
>
>   - 1. On the claim that our improvements stem mainly from ''DDPM schedules'':
>
>       We believe part of the reviewer’s interpretation may stem from a **misunderstanding**, and we clarify below. We would like to clarify that the backbones showing the strongest gains in our experiments **do not use the original DDPM linear schedule**. For example, EDM explicitly designs a new molecular noise schedule, inspired by but distinct from the cosine schedule proposed in iDDPM. Thus, even schedules that have already been optimized for molecular data (and differ substantially from classic DDPM) still contain non-negligible redundant high-noise segments, and our method continues to offer both efficiency and accuracy improvements on top of them.

---

> > ### Author Response · Authors · 2025-11-19
> > **Response to Reviewer FhFk - Part II**
> >
> > - 2. On models that explicitly address schedule redundancy (like GeoBFN, SLDM, EquiFM, etc.)
> >
> >       We appreciate the reviewer for highlighting these excellent developments, and we have now incorporated all of them into the Related Work section. At the same time, several clarifications are important:
> >
> >       **GeoBFN uses atom charges rather than one-hot atom types.** Because these cannot be zero-centered without altering semantics, the zero-mean invariance required for our GA analysis does not hold. Hence, GeoBFN lies outside the scope of data modalities to which our method applies.
> >
> >       SLDM is a very recent architecture with significantly different training formulation, and **we have included it in the discussion.** We would also like to note that SLDM is accepted by NeurIPS 2025, which hasn't be held now, so we could only read the arXiv version. According to ICLR policy[1], authors are not required to compare against papers available solely on arXiv. We hope this won’t be seen as a downside of our work.
> >
> >       EquiFM has similar effect to GeoBFN and SLDM with much less steps in generaion. And it was included in our experiments. **Even in the EquiFM setting, our method further reduces the path from 200 steps to 160 steps while still improving generation quality.** This demonstrates that even after aggressive schedule optimization, forward trajectories retain a residual Gaussian regime that can be truncated analytically.
> >
> >       These observations suggest that schedule optimization alone does not fully eliminate redundant over-noised regions, and that our Gaussian approximation–based criterion can still detect and remove additional unnecessary segments. In the revision, we include more detailed discussion about the connection with these works on schedule optimization.
> >
> >     [1] ICLR Policy: https://iclr.cc/Conferences/2026/ReviewerGuide
> >
> > - W3: Gaps in research logic and writing:
> >
> >   - 1. Cumulative tensors, mutual information, and cumulative distribution functions are used but the theoretical analysis relies on the first metric.
> >
> >     - We think issue arises from a **misunderstanding**, and we clarify it explicitly in the revised manuscript.
> >     The cumulant-based functional in Eq. 8 is **only a theoretical Gaussianity measure**, introduced to support Proposition 3.1 and to reason about how initial non-Gaussianity influences the ordering of $T^\ast$ across datasets. Based on the theoretical measure, we utilize the MI-based dependency score  and the KS-based CDF distance as the **practical operational metrics** to estimate $T^\ast$ empirically. Each captures a different and necessary aspect of Gaussianity, i.e., the data dependency and distributional similarity. **Using only one of these metrics would be insufficient: low MI does not guarantee Gaussian marginals, and low KS distance does not guarantee independence.** Together, they provide the complementary checks needed to validate Gaussian Approximation in practice.
> >     In the revision, we make this explicit in Sec. 3.3 to emphasize that **the paper uses dependency measurement and distributional similarity criteria, rather than three unrelated metrics.**
> >
> >   - 2. Lacks of details in the introduction of MI test.
> >
> >     - **In the revision, we have expanded the explanation of Eq. (26) with the structure of $\mathbf{x}_t$ in molecular dataset as an example.**
> >
> >       For all GPPGMs considered, the noised molecular data $\mathbf{x}_t$ is represented as a matrix of shape $\mathbf{x}_t \in \mathbb{R}^{N \times d}$ where $N$ is the number of components (atoms) and $d$ is the feature dimension (3D coordinates and atom-type features). Importantly, the MI test is applied **directly to the raw model inputs**, not to learned neural features, ensuring that our Gaussianity evaluation is fully model-agnostic. Because a valid Gaussian Approximation requires that all dimensions of $\mathbf{x}_t$ behave as **independent** Gaussian variables, so we require that both **feature-wise and component-wise** MI must fall below the tolerance threshold.
> >
> >       To improve the clarity and the consistency with Appendix. D, in the revised manuscript, we modify the notations of $X_{rows}$ and $X_{cols}$ to $X_{feat}$ and $X_{comp}$, respectively. They represent the MI across different dimensions of $\mathbf{x}_t$, i.e., from the feature-wise (across different atomic features) or the component-wise (across different atoms). These clarifications have been incorporated into the Appendix F.1 of the revised manuscript. Thanks for the reviewer's kind reminders.

---

> > > ### Author Response · Authors · 2025-11-19
> > > **Response to Reviewer FhFk - Part III**
> > >
> > > - 3. Unclear visualization of Gaussianity in Figure 3.
> > >
> > >     - In the revision, we made two improvements to clarify the notion of ''Gaussianity'' in Figure 3:
> > >      1. **We added a formal definition of Gaussianity.**
> > >      We now explicitly define Gaussianity as the degree to which a dataset or random variable aligns with a Gaussian distribution. **In Appendix D, we also quantify the Gaussianity of different intial data distributions** using the same two criteria employed in our GA timestep estimation, **dependency decay** (via MI) and **distributional similarity** (via KS test). This provides a principled numerical characterization that complements the visualization.
> > >
> > >      2. **Regarding numerical values in the figure:**
> > >      While numerical Gaussianity metrics are now included in Appendix D, **it remains challenging to summarize the full Gaussianity of a dataset using a single scalar in the figure**, as Gaussianity arises from multiple statistical aspects (e.g., independence and marginal alignment). Therefore, we keep the color-based visualization in Figure 3 for clarity and readability, but we have **updated the caption to reference the corresponding quantitative Gaussianity analysis provided in Appendix D.**
> > >
> > >    We believe this combination of formal definition, quantitative metrics, and visual illustration provides a more complete and concrete presentation of Gaussianity in the revised manuscript.
> > >
> > > ### Questions
> > >
> > > - Q1: Zero-mean assumption is a major limitation of your method.
> > >
> > >   - We agree that **the zero-mean assumption currently limits the direct applicability of our method, and we explicitly acknowledged this limitation in the Sec. 6.** However, we believe extending our method to non–zero-mean distributions is a promising next step. One potential idea we are already exploring is to treat the global mean as a **sampled variable**, for instance, sampling it from the empirical distribution of dataset means, so that the Gaussian approximation can incorporate both mean and covariance while preserving the original data semantics. Such an approach would generalize our framework beyond zero-mean–invariant domains.
> > >
> > > - Q2: The performance if timestep truncation is applied only during sampling.
> > >   - This is an interesting question. To isolate the effect of truncation at **sampling only**, we conducted an additional experiment on EDM where we apply our Gaussian Approximation (GA) using the **official pretrained weights**, without retraining the model on the truncated trajectory.
> > >
> > >     Firstly, we would like to claim that this setting is **not well aligned** with our frameworkfrom a theoretical perspective: the original EDM operates on **non–zero-mean one-hot atom-type features**, whereas our GA derivation assumes **zero-mean invariance**. Projecting the high-noise states onto a zero-mean Gaussian manifold therefore introduces a systematic deviation of the generative trajectory from the original probability path. This mismatch can in principle **harm generation quality**, even if the empirical degradation is small or dataset-dependent.
> > >
> > >     For clarity, we compare the EDM + GAGA setting in the main paper (where GA is applied consistently in both training and sampling under zero-mean preprocessing) with the “sampling-only GA on official weights’’ variant:
> > >
> > >     | Method                                        | Atom Sta (%) | Mol Sta (%) | Valid (%) | Valid * Uniq (%) |
> > >     |-----------------------------------------------|--------------|-------------|-----------|------------------|
> > >     | **EDM + GAGA (training + sampling, ours)**    | 98.9         | 85.6        | 94.7      | 92.0             |
> > >     | **EDM + GAGA (sampling only, official weights)**| 88.1            | 26.6           | 41.8         | 39.2                |
> > >
> > >     Here, the first row corresponds to the principled GAGA pipeline used in the main paper, where the model is trained and sampled on a consistently truncated, zero-mean trajectory. The second row represents the ablated setting requested by the reviewer, where **GA is applied only at sampling time on the original EDM weights**; we emphasize that this variant should be interpreted as an empirical sanity check rather than the recommended or theoretically justified usage of our method. **It can be seen that the GA with estimated steps applied on non-zero meaned data will cause severe harm to generation quality.**

---

> > > > ### Author Response · Authors · 2025-11-19
> > > > **Response to Reviewer FhFk - Part IV**
> > > >
> > > > - Q3: A curious question: for element types alone, is their distribution closer to a Gaussian than that of images?
> > > >   - Our response to this question has two parts.
> > > >
> > > >      **1. Empirical comparison of full molecular data vs. images.**
> > > >      In Appendix D of the revised version, we include a quantitative comparison between **QM9 molecular data** and **CIFAR-10 images** using both KS test and MI-based dependency measures. These results clearly show that the *overall* molecular data distribution is closer to a Gaussian than image data. This supports our main claim that molecular modalities enter the Gaussian regime much earlier.
> > > >
> > > >      **2. Regarding “element types alone.”**
> > > >      Conceptually, we do not encourage comparing **only the one-hot atom-type features** with images because the forward process corrupts *the entire feature vector jointly*, not position and atom types separately. Gaussianity should therefore be understood at the level of the **full joint distribution**, not individual feature subsets.
> > > >
> > > >      Nevertheless, for completeness, we performed the requested comparison by isolating only: the **one-hot atom-type** features from QM9, and the **normalized pixel intensities** from CIFAR-10.
> > > >
> > > >      We report KS p-values and MI scores below:
> > > >
> > > >      | Dataset                | KS\_p\_value      | MI_total (feature-wise + components-wise)         |
> > > >      |------------------------|-------------------|----------------------------------|
> > > >      | **QM9\_atom\_types**   | 3.95e-04          | 0.1836485         |
> > > >      | **CIFAR-10 (pixels)**  | 4.13e-03          | 1.79879646          |
> > > >
> > > >      **Even when considering atom types alone, the dependency structure (MI) is substantially lower than that of CIFAR-10 pixels, consistent with the intuition that one-hot categorical molecular features are much sparser and less spatially correlated than image intensities.**
> > > >
> > > >      We have clarified the overall similarity to Gaussian and provided the quantitative results about Gaussianity of initial data distributions in the revised manuscript.
> > > >
> > > >
> > > >   We hope that our obove clarifications could help resolve the concerns raised, including those that may have arisen from misunderstandings. We would greatly appreciate your reconsideration of the evaluation, and we are happy to provide any additional information if needed.

---

> ### Comment · Reviewer_FhFk · 2025-11-28
>
> Thank you for your response. The manuscript’s clarity has improved significantly, and my questions have been mostly addressed. While the proposed method seems straightforward, the method’s good performance and the valuable Gaussianity observation for molecular data benefit the community. So I’m pleased to increase the score to 6. ( It seems I cannot edit the score now, I will update once the system is fixed.)

---

### Official Review · Reviewer_GRXx · 2025-10-31

**Soundness:** 3
**Presentation:** 1
**Contribution:** 2
**Rating:** 2
**Confidence:** 4

**Summary:**

This paper proposes an algorithm to calibrate the noise schedule for diffusion model training and sampling. The central idea is to identify a minimal log signal-to-noise ratio (log-SNR), $\lambda_\min$ (in the sense of [1]), at which the noised latent variable attains sufficient Gaussianity. The schedule is then truncated at this point to avoid unnecessary and potentially harmful levels of noising. The effectiveness of this calibration method is demonstrated on several diffusion models for molecule generation.

[1] Kingma, Diederik, and Ruiqi Gao. "Understanding diffusion objectives as the elbo with simple data augmentation." Advances in Neural Information Processing Systems 36 (2023): 65484-65516.

**Strengths:**

- Clear Motivation: The paper is well-motivated, addressing the common and practical problem of inefficiently wide noise schedules in diffusion models.
- Strong Empirical Support: A key strength is the empirical demonstration that baseline models often operate on an unnecessarily broad SNR range. The results convincingly support the claim that truncating this range to an "effective" one can be done without degrading model performance.
- Practical Significance: The proposed method offers a practical tool for scheduler calibration. A significant benefit is the potential to accelerate inference by reducing the number of sampling steps, even when applied post-hoc to models trained with a standard, non-calibrated scheduler.

**Weaknesses:**

- Insufficient Positioning: A significant weakness is the lack of thorough positioning against the extensive existing literature on noise schedulers, SNR analysis, and related diffusion model theory (e.g., [1,2,3]). This omission causes the work to feel disconnected from established formalism in the area. Consequently, the development of the method appears somewhat ad-hoc rather than being rigorously derived from first principles.
- Clarity and Readability: The paper's clarity could be improved. A considerable portion of the text is dedicated to standard, boilerplate descriptions of diffusion models, which detracts from the space available to elaborate on the novel contributions.
- Undefined Key Terminology: A central term, "data identity," is used repeatedly throughout the manuscript but is never formally defined. This ambiguity leaves a core concept of the paper open to interpretation and hinders the reader's ability to fully grasp the method.
- Unconventional Presentation: In a minor editorial point, the results tables use boldface to highlight the proposed method's results rather than the conventional practice of bolding the best-performing method for a given metric. This unconventional choice makes direct performance comparisons less intuitive.

[1] Kingma, Diederik, and Ruiqi Gao. "Understanding diffusion objectives as the elbo with simple data augmentation." Advances in Neural Information Processing Systems 36 (2023): 65484-65516.

[2] Falck, Fabian, et al. "A Fourier Space Perspective on Diffusion Models." arXiv preprint arXiv:2505.11278 (2025).

[3] Gao, Ruiqi, et al. "Diffusion models and gaussian flow matching: Two sides of the same coin." The Fourth Blogpost Track at ICLR 2025. 2025.

**Questions:**

1. Could the authors please provide a formal, mathematical definition for the term "data identity" as it is used in the context of this paper?
2. The paper mentions benefits for training. Could the authors clarify precisely how the proposed calibration method reduces training time?

---

> ### Author Response · Authors · 2025-11-19
> **Response to Reviewer GRXx - Part I**
>
> Thank you very much for your valuable feedback! We revise our manuscript accordingly and provide our responses below.
>
> ### Weaknesses
>
> - W1: Insufficient Positioning: lack of thorough positioning against existing literature on noise schedulers, SNR analysis, and related diffusion model theory.
>
>   - We thank the reviewer for highlighting the importance of positioning our method within the broader literature on noise schedules, SNR analysis, and diffusion-theoretic foundations. **We agree that these works share conceptual connections with ours, and we have expanded the Related Work section accordingly to provide clearer context.**
>
>     We agree that analyses of noise schedules and SNR behavior reveal essential properties of Gaussian probability paths, including how data statistics evolve during the forward noising process. However, we would like to emphasize an important distinction. Prior works primarily investigate how to design or analyze forward schedules, often proposing alternative SNR curves or theoretical characterizations. **In contrast, our objective is orthogonal: we aim to accelerate a pre-defined GPPGM (e.g., diffusion or flow matching models) without modifying its schedule or altering its training dynamics.** For instance, EDM[1] explored the effect of noiseschedules for molecular generation and thus proposed their new schedule for molecular generation. However, our method can still be directly applied as a plug-in to shorten the trajectory while preserving model fidelity.
>
>     Our contribution is thus complementary: **it does not propose a new schedule, nor does it require changing the predefined forward process or the generation trajectory granularity; instead, we provide a principled criterion to identify when data identity is sufficiently lost so that the remaining trajectory can be replaced by an analytically tractable Gaussian.** It shares some similar insights with SNR analysis. However, our method provides a principled method to truncate the pre-defined probabilistic path. This truncation is dependent on both forward schedule and intial data distribution, which can not be directly acheived by SNR analysis. Moreover, our method allows us to retain the full resolution of the original generative paths, which may have differnt predefined settings across different models. In the revised manuscript, **we have added a new paragraph in Sec. 5 to clarify these conceptual differences and to better articulate how our contribution fits within the established literature.** We hope this additional discussion resolves this concern and makes the positioning of our work clearer.
>
>     [1]Hoogeboom, Emiel, et al. "Equivariant diffusion for molecule generation in 3d." International conference on machine learning. PMLR, 2022.
>
> - W2: Clarity and Readability: boilerplate descriptions of diffusion models.
>
>   - We agree that excessive introductory material can distract from the core methodological contributions, and **we have revised the manuscript accordingly.**
>
>     First, we reduced and streamlined the boilerplate exposition of the forward process in GPPGMs. **The revised version focuses only on the essential components needed to motivate our approach, allowing more space to highlight our key ideas and empirical findings.**
>
>     Second, in response to both this comment and Weakness 3, **we refined the conceptual presentation by replacing the original notion of data identity with Gaussianity.** Since identity decay and Gaussianity increase are equivalent under the forward Gaussian probability path, introducing the method directly through the lens of Gaussianity provides a clearer and more intuitive narrative. **This revision improves conceptual coherence and reduces the cognitive load on the reader, making the contribution easier to follow without changing any underlying technical content.**
>
>     We believe these adjustments meaningfully enhance readability and help ensure that the novel aspects of our method are communicated more directly and effectively.

---

> > ### Author Response · Authors · 2025-11-19
> > **Response to Reviewer GRXx - Part II**
> >
> > - W3: Undefined Key Terminology: ''data identity'' is not defined.
> >
> >   - We agree that the previous use of the term data identity was insufficiently formalized and could cause confusion. To address this, we made two major revisions in the updated manuscript:
> >
> >     1. **We replaced the term “data identity’’ with the more precise and formally grounded concept of Gaussianity.** In our original manuscript, the notion of identity loss is equivalent to the point where the noised data distribution becomes sufficiently close to a Gaussian distribution. Therefore, expressing our method directly in terms of Gaussianity is both clearer and more aligned with the underlying theory.
> >
> >      2. **We restructured the exposition around explicit, quantitative criteria: Dependency decay measured via mutual information, and Distributional similarity measured via a KS-based Gaussianity test.** These tools now provide a rigorous and operational definition of the condition when the corrupted distribution enters the Gaussian regime required by our approximation.
> >
> >     By grounding the discussion in the measurable property of Gaussianity rather than the previously informal term identity, we believe the description of our method is now clearer, more intuitive, and easier for readers to interpret without ambiguity. Again, we thanks for the suggestion from the reviewer.
> >
> > - W4: Unconventional Presentation of Bold Words.
> >
> >   - We agree that boldfacing the best-performing entry is the conventional practice. In our case, however, **our method is designed as a plug-in module that can be applied to a variety of existing GPPGM backbones without modifying their architectures, noise schedules, or training setups.** Consequently, the primary purpose of our tables is not to claim a new state-of-the-art model, but to **highlight the consistent improvement** that our method brings across diverse baselines. For this reason, we boldfaced each “baseline + our method’’ row to make the improvement over its corresponding backbone immediately visible. To reduce the unconventionality, **in the revision, we follow GeoLDM[1] to use the grey background to highlight our method instead of the bold words.** We appreciate the reviewer’s concern about readability. In the revised version, we replace the boldface representation with grey background, which is convention in previous works.
> >
> >     [1] Xu, Minkai, et al. "Geometric latent diffusion models for 3d molecule generation." International Conference on Machine Learning. PMLR, 2023.
> >
> > ### Questions
> >
> > - Q1: A formal, mathematical definition for "data identity".
> >
> >   - We agree that the previous use of the term data identity was insufficiently formalized and could cause confusion. In our original manuscript, the notion of identity loss is equivalent to the point where the noised data distribution becomes sufficiently close to a Gaussian distribution. Specifically, the **Gaussianity is defined as a dataset or random variable distributed according to a Gaussian distribution.** Consequently, in the new version of the manuscript, **we replaced the term “data identity’’ with the more precise and direct word, Gaussianity, which is directly evaluated in our method for GA.** In such case, the proposition 3.1 will directly serve for the relationship between Gaussianity of initial data distribution and the GA timestep $T^*$.
> >
> > - Q2: Details about benefits in training.
> >
> >   - As noted, the revised manuscript now uses the formally grounded notion of *Gaussianity*, which is measured through both dependency decay and distributional similarity. **As for the implementation details for training, we now provide more in Appendix G.2.** For example, in EDM, the original configuration trains 1000 forward steps for 3000 epochs on QM9. After identifying a truncation at $T^* = 550$, the maximum timestep during training is reduced to 550. To preserve comparable training sufficiency, we **proportionally scale** the number of epochs from the original setting ($3000$) to $3000 \times \frac{550}{1000} = 1650$ epochs. Under this consistent and fair setting with fewer training steps and fewer sampled timesteps, our plug-in method still enables each baseline to outperform its original version, while providing substantially more efficient training and sampling.
> >
> > We believe our responses clarify the issues raised, and we further improved our presentation to make the manuscript more readable. We would be grateful if the reviewer could reconsider the assessment, and we welcome any additional questions.

---

### Official Review · Reviewer_1nJS · 2025-10-31

**Soundness:** 2
**Presentation:** 3
**Contribution:** 3
**Rating:** 4
**Confidence:** 4

**Summary:**

This paper proposes an interesting approach for accelerating the training and sampling procedures of 3D molecular diffusion and flow-based models using Gaussian Approximation. The key idea is that molecular data becomes "sufficiently Gaussian" early on in the forward noising process; therefore, the trajectory can be truncated and replaced by closed-form Gaussian distributions. Experiments yield promising results in terms of improved sampling and training efficiency, as well as enhanced generation quality, on two 3D molecular datasets.

**Strengths:**

* The idea of assessing the Gaussianity of the data during the noising process to accelerate the training and sampling is novel and sound. Additionally, leveraging this specifically for molecular data is interesting (although some claims in this regard require further clarification, as noted in the weaknesses).
* The paper is well-written and clearly explained.
* The experiments covering two common datasets and a few baselines are rather extensive and show promising results.

**Weaknesses:**

* The main assumption in this paper is that molecular data converges to Gaussianity faster than other modalities, e.g., images. However, this assumption is neither theoretically justified nor empirically verified. Proposition 3.1 requires that the initial distribution is more Gaussian; however, can this be demonstrated? Additionally, based on Figure 2, the authors claim that image data retains recognizable features for more steps than molecular data; however, this can be misleading, as images are much more familiar to the human eye than molecules. Quantitative or qualitative results are needed to verify this claim. For instance, it would be interesting to plot the proposed dependency evaluator $Dep(x_t)$ and the distributinal similarity $D_t$ over different timesteps during the noising process for an image dataset and a molecular dataset (of course, they should be adequately scaled to accommodate the different data scales and ranges).
* While the paper does a good job at explaining how $T^* $ is obtained, it is missing some details on how the training and sampling procedure is adapted. It seems that the segment $[T^*, T]$ is simply truncated; however, the paper should formally clarify this. For example, what are the exact modifications to the training and sampling algorithms from EDM [1]?
* All reported experimental results lack standard deviations across different runs, which makes it hard to assess the statistical significance of the claimed results. Also, the code is not provided.
* It would be a great addition to the paper if the proposed Gaussianity test is used in other settings beyond accelerating training and sampling procedures. In some cases, efficiency is not the primary concern, and the user may be willing to sacrifice efficiency for improved generation quality. I believe the proposed approach can be leveraged in this setting. For example, given a pre-specified noise schedule, find the total time steps $T$ such that the optimal $T^* =1000$ or use the proposed Gaussianity test to guide the design of noising schedules such that $T^*=T$. This setting would enable comparison with baselines for the same computational budget, potentially yielding improved generation results because of better allocation of time steps.

I would be willing to increase my score if the authors address these points convincingly.

**References**

[1] Hoogeboom, Emiel, et al. "Equivariant diffusion for molecule generation in 3d." International conference on machine learning. PMLR, 2022.

**Questions:**

1. Is the term "Gaussian Probability Path based Generative Models (GPPGMs)" known in the literature? If yes, please cite the corresponding papers. If not, please somehow clarify this or use more well-known terms.
2. Figure 1 is not clear. What is "complex", and what is the difference between the two backward passes? Please clarify the figure and the caption.
3. In Equation 2, $\alpha_{t-\Delta t|t}$ is not defined.
4. In Equation 7, why is $\tilde v_t$ chosen in this way? Please clearly explain this.
5. In Equation 10, shouldn't $m>=3$?
6. In line 225, it is stated that "sparse molecular coordinates around equilibrium are closer to Gaussian". Please elaborate and give concrete qualitative or quantitative arguments.
7. When estimating all the quantities required for computing $T^*$, e.g., the variance of the Gaussian, the dependency evaluator, and the distributional similarity, do you sample multiple noise vectors for each data sample and each noise level?
8. What are the values of $\epsilon_{dep}$ and $\epsilon_{DS}$ and how are they chosen?

---

> ### Author Response · Authors · 2025-11-19
> **Response to Reviewer 1nJS - Part I**
>
> Thanks for your valuable feedback, we have revised our manuscript and we provide some responses below:
> ### Weaknesses
> - W1: Demonstration of different Gaussianity across different data distributions. Quantitative or qualitative results to verify the claim about different timesteps when obtaining sufficient Gaussianity.
>
>   - Proof of initial Gaussianity difference: Indeed, Proposition 3.1 is shown to demonstrate the relationship between initial distribution and GA timestep $T^*$. Based on this point, we found that the initial molecular data distribution is closer to Gaussian, **we provide the empirical results below**:
>
>     | Dataset      | KS-p-value                       | MI_total (feature-wise + component-wise)                       |
>     | ------------ | -------------------------------------------- | ------------------------------------------- |
>     | **QM9**      | $8.83\times10^{-8} \pm 1.43\times10^{-11}$ | $3.34\times10^{-1} \pm 1.39\times10^{-3}$ |
>     | **CIFAR-10** | $2.32\times10^{-3} \pm 7.00\times10^{-4}$  |     $1.80\times10^{0} \pm 3.60\times10^{-1}$  |
>
>     To ensure the consistency of Gaussianity evaluation, we use the same metric for initial distributions' comparison and GA timestep estimation. This experiments is constructed on 10000 samples from QM9 dataset (molecules) and Cifar-10 dataset (images). It shows that for the initial data, from both perspectives of data dependency and distributional similarity, **the molecular data from QM9 dataset is closer to Gaussian compared with image data from Cifar-10 dataset**. This results is also shown in the Appendix D of the revised manuscript.
>
>   - Quantitative verification of Figure 2: In Figure 2, we claim that the image data retains more identity compared with molecules, because in this paper, the identity is directly corresponding to Gaussianity. In the new version of the manuscript, we have modified the claim of identity to Gaussianity directly. Moreover, it is true that the visualized atom distribution is less familar to human eyes. However, throught the row (b), i.e., the distribution of atom types. We can clealy observe that at very early steps, **the atom types have been drastically changed**. This leads to the directly change for molecules, from one stable molecule to another unstable, or even invalid molecule. In such case, the noised data is no more preserving the identity of the orginal molecule. We agree that the quantitative experiments may express more information, **so in the revision, we report the quantitative Gaussianity across initial data distribution in Appendix D, and based on the Prop. 3.1, it can be analytically proved that the sufficient Gaussainity will be obtained ealier on molecular data, while the image data will obtain it later.**
>
>
> - W2: Details on how the training and sampling procedure is adapted?
>
>   - **Now we provide more details about applying our method on baselines in Appendix F.2.** For example, in EDM, the original configuration trains 1000 forward steps for 3000 epochs on QM9. After identifying a truncation at $T^* = 550$, the maximum timestep during training is reduced to 550. To preserve comparable training sufficiency, we **proportionally scale** the number of epochs from the original setting (3000) to $3000 \times \frac{550}{1000} = 1650$ epochs. In sampling, the start point will also be modified to 550 timestep with the analytically derived Gaussian. Consequently, the sampling will also start from $550$ steps with the analytically derived Gaussian. Under this consistent and fair setting with fewer training steps and fewer sampled timesteps, our plug-in method still enables each baseline to outperform its original version, while providing substantially more efficient training and sampling.
>
>
> - W3: Lack of standard deviations and code.
>
>   - **Now we also report the standard deviation in the comparison with baselines. The code is now available at:** https://anonymous.4open.science/r/GAGA-Anonymous-1375. Specifically, we provide an illustrative example based on EDM to intuitively demonstrate the workflow of our method when applied to a constructed baseline. The reader could then easily run our method on EDM to verify the performance. We’d also like to share that ICLR does not require authors to release code at submission[1], and we hope this won’t be seen as a downside of our work.
>
>     [1] ICLR 2026 Author Guide: https://iclr.cc/Conferences/2026/AuthorGuide

---

> > ### Author Response · Authors · 2025-11-19
> > **Response to Reviewer 1nJS - Part II**
> >
> > - W4: Potential extension to improving the performance instead of efficiency.
> >
> >   - We thank the reviewer for this interesting suggestion. Indeed, the proposed Gaussianity test can, in principle, be used not only for accelerating generation but also for **redistributing** timesteps when efficiency is not the primary concern.
> >
> >     Intuitively, given a pre-specified noise schedule with total steps $T$, one could enlarge $T$ (for example, increasing EDM from $T=1000$ to $T=5500$) and then apply our Gaussianity test to locate the corresponding Gaussianity timestep $T^* \approx 1000$. This would force a larger proportion of timesteps to lie in the identity-preserving region of the trajectory, potentially improving generative quality by providing finer resolution where it matters most. Similarly, one could design schedules such that $T^* = T$, ensuring that all computation is allocated to the non-Gaussian region.
> >
> >     However, this setting introduces a **confounding effect**: increasing $T$ also increases the **granularity of the trajectory itself**. As a result, if generation quality improves, it becomes difficult to determine whether the improvement arises from the GA or simply from the substantially finer discretization. In such cases, we cannot cleanly attribute the benefit to our method.
> >
> >     For this reason, our paper focuses on the regime where the computational budget is fixed or reduced, and we use the Gaussianity test to **remove** redundant timesteps rather than redistribute or increase them. We agree that extending the Gaussianity test for step reallocation at controlled compute is an interesting future direction, and **we include this discussion as the futural direction in the revision.**
> >
> > ### Questions
> > - Q1: Clarification of GPPGMs.
> >   - We follow the definition of Gaussian probability paths introduced in [1], which shows that “Flow Matching is compatible with **a general family of Gaussian probability paths for transforming between noise and data samples**—subsuming existing diffusion paths as specific instances.” Motivated by this unifying perspective, we adopt the term Gaussian Probability Path based Generative Models (GPPGMs) to collectively describe generative models such as diffusion models and flow matching models, both of which operate along Gaussian probability paths.
> >
> >     [1] Lipman, Yaron, et al. Flow Matching for Generative Modeling. arXiv:2210.02747, 2022.
> >
> > - Q2: Clarification of 'Complex' in Figure 1.
> >   - In Figure 1, the complex data distribution represents those distributions can not be zero-meaned, like image. In the revision, **We have replaced the 'complex' with 'Non zero-meaned' in the figure for better clarity.**
> >
> > - Q3: Clarification of Equation 2.
> >   - To improved the readibility and simplify the notations, **we updated the Equation (2) in the revision to a more readable form.**
> >
> > - Q4: Clarification of $\tilde{v}_t$ in Equation 7 (Equation 6 in the revision).
> >   - In Equation 6, we firstly estimated the per-sample variance of dataset and use it as the variance of $x_0$, then we **plug in it to the VP noising process, as shown in Equation (1), to get the per-sample variance at timestep $T$**. This estimated variance will finally used in Gaussian Approximation. To make it clearer, we add some explanations in the new version of the manuscript:
> >   "Therefore, under the VP forward process on zero-meaned data, we plug in $\hat{v}$ into the Equation 1 as the variance of data samples, then the mean $\tilde{\mu}_t$ and variance $\tilde{v}_t$ of noised data at intermediate timesteps $t$ can have the following analytic form:"
> >
> > - Q5: In Equation 10 (Equation 9 in the revision), shouldn't $m \ge 3$?
> >   - In Equation 9, we actually intend the condition to **hold for all $m = 2,\dots,K$**, including the second-order term. To make this clear, in the revision **we optimize the Equation 7 with an indicator function.** With this clarification,Equation 9 is well defined for $m = 2,\dots,K$: the case $m = 2$ only uses the covariance term, while orders $m > 2$ add higher-order cumulants. We update the text around Prop. 3.1 accordingly.
> >
> > - Q6: Quantitative elaboration of ''sparse molecular coordinates around equilibrium are closer to Gaussian.''.
> >   - In the revision, to support this claim quantitatively, **we now provide an explicit comparison between image and molecular data in Appendix D using exactly the two Gaussianity criteria proposed in our paper**: (i) dependency-based measures and (ii) distributional similarity. Concretely, we compute feature-wise and component-wise mutual information, as well as p-value of KS test, for the *initial* distributions of image dataset and molecular datasets. **The results show that the initial molecular data exhibit consistently lower dependency and smaller deviations from a fitted Gaussian than the image data, indicating that they are empirically “closer to Gaussian’’ under our metrics.**

---

> > > ### Author Response · Authors · 2025-11-19
> > > **Response to Reviewer 1nJS - Part III**
> > >
> > > - Q7: Details about sampling noise vectors in statistical tests?
> > >   - We thank the reviewer for raising this important implementation question. In practice, for each data sample at a specific noise level, we **sample one noise vector**. **Moreover, the Gaussianity evaluation is performed heuristically, and each tested timestep uses an independently sampled noise vector.** This results in a diverse collection of noisy observations across the trajectory, closely mirroring the standard GPPGM training procedure, where noise and timestep are repeatedly resampled. Because our statistics are computed across sufficiently large batches and multiple timesteps, this provides stable and reliable estimates without needing to sample multiple noise vectors per data point at each level. **In the revision, we include this statement in Appendix F.**
> > >
> > > - Q8: What are the values of $\epsilon_{\mathrm{dep}}$ and $\epsilon_{\mathrm{DS}}$ and how are they chosen?
> > >   - In all our experiments, we set $\epsilon_{\mathrm{dep}} = 0.1$, meaning that **the mutual information score must fall below $0.1$** to be considered sufficiently independent. For the distributional similarity threshold, **we set the confidence of KS-test as 95%, and then derive $\epsilon_{\mathrm{DS}}$ accordingly. These values are now explicitly documented and discussed in Appendix F.2.**
> > >
> > > We believe our responses clarify the issues raised. We would be grateful if the reviewer could reconsider the assessment, and we welcome any additional questions.

---

### Author Response · Authors · 2025-11-20
**Public Response About Manuscript Revision**

We would like to thank all reviewers for their constructive feedback and, in particular, Reviewer GRXx for the helpful suggestion regarding the terminology ''identity'' used in our manuscript. Following this advice, **we have replaced the term ''identity'' with the more precise and directly interpretable notion of ''Gaussianity''**. In our original submission, the timestep at which the data “loses its identity” is exactly the timestep at which it attains sufficient Gaussianity; we agree that expressing this concept explicitly in terms of Gaussianity leads to clearer presentation and improved readability.

To reflect this clarification, we have updated both the title and the manuscript accordingly. We believe these changes make the contribution more accessible and faithfully aligned with the underlying technical insight.

We sincerely appreciate the reviewers’ and AC’s time and thoughtful comments, and we look forward to hearing your further feedback.

---

### Author Response · Authors · 2025-11-27
**Kind Reminder About Our Rebuttal**

Dear Reviewers,

We would like to kindly follow up on our rebuttal. **It has now been over a week since we submitted our responses, and only one week remains before the discussion period ends.** We believe we have thoroughly addressed all concerns through our revisions and rebuttal, and we would deeply appreciate hearing your further thoughts.

We also kindly hope that you may raise your ratings in light of the clarifications and improvements we have provided. Your additional feedback would be invaluable for strengthening the final version of the work.

Thank you again for your time and for engaging with our submission.

Sincerely,

Authors

---

### Meta-Review · Area_Chair_Sqbx · 2026-01-04

**Summary:**

Reviewer 1nJS raised a set of constructive concerns primarily centered on methodological justification, clarity, and empirical completeness rather than fundamental flaws in the proposed approach. Specifically, the reviewer questioned: (1) insufficient justification for the Gaussianity assumption, (2) unclear adaptation of training and sampling under trajectory truncation at T*, (3) missing standard deviations and unclear code availability, (4) notation and clarity issues, and (5) potential broader applicability of Gaussianity tests.

Reviewer GRXx raised concerns about (1) positioning and novelty relative to prior diffusion literature, (2) clarity and readability of the presentation, (3) ambiguous or informal terminology, and (4) insufficient explanation of practical training-time benefits.

Reviewer FhFk raised five main concerns. (1) the reviewer questioned whether trajectory truncation is genuinely distinct from prior schedule-modification approaches, and whether the Gaussian approximation used in truncation is theoretically consistent or introduces additional tunable hyperparameters, (2) unclear gains for models with optimized schedules, (3) gaps in logic and writing, (4) zero-mean assumption may limit the generality, and (5) clarification needed on sampling-only truncation and atom-type–only Gaussianity.

**Reviewer Concerns:**

The authors explained and solved all concerns raised by the reviewers by providing additional experimental results and revising the manuscript accordingly.

**Reviewer Scores:**

For Reviewer 1nJS: Based on the rebuttal and subsequent revisions, particularly the addition of quantitative Gaussianity comparisons, detailed clarification of the truncated training and sampling procedure, improved experimental reporting with standard deviations, clarified notation and methodological definitions, and the release of an anonymous code repository, the reviewer’s concerns appear to have been fully addressed. Given the reviewer’s expressed willingness to revise their evaluation contingent on these points, it is reasonable to expect that the reviewer would increase their score to 6 (marginally above the acceptance threshold), reflecting improved confidence in the empirical grounding, clarity, and reproducibility of the work.

Reviewer GRXx would increase the rating to 6 (marginally above the acceptance threshold) given that the the revision and rebuttal have addressed the major concerns.

As to Reviewer FhFk's concerns, the authors clarified all of them and provided additional supporting experiments, the Reviewer FhFk expressed satisfaction with authors' point-to-point solutions and increased the rating to 6.

---

### Decision · Program_Chairs · 2026-01-26

Accept (Poster)

---

> ### Comment · Reviewer_GRXx · 2026-03-09
> **Formal Concern: Review Integrity and Decision Irregularity for Submission 2189**
>
> Dear Program Chairs and Senior Area Chairs,
>
> I am writing to you today to express my serious concerns regarding the recent acceptance decision for submission 2189.
>
> My motivation for reaching out stems from a deep commitment to the integrity of our community’s peer-review process. Having observed several significant irregularities in how this specific case was handled by the Area Chair (AC), I feel it is my professional responsibility to bring these matters to your attention.
>
> The decision to accept this paper is concerning for the following reasons:
>
> - Discrepancy Between Scores and Outcome: The submission received two "Strong Reject" ratings and one "Weak Reject." An acceptance under such a consensus is highly irregular and, in my view, undermines the efforts of the reviewers who identified fundamental flaws in the work.
> - Violation of Official Guidelines: I noted that the AC’s meta-review explicitly took into account a reviewer’s willingness to raise their score post-rebuttal. This directly contradicts the official guidelines provided to us, which stated that such shifts should be ignored due to the recent OpenReview leakage.
> - Unfounded Assumptions of Reviewer Agreement: Most distressingly, the AC assumed in their justification that the other two reviewers—myself included—were satisfied with the authors' rebuttal and would likely move from a "Strong Reject" to a "Weak Accept." This assumption was made without any consultation.
> - Scientific Dishonesty in the Rebuttal: At the time commenting was blocked, I was preparing a response to refute the authors’ claims. While the authors cited the related work I suggested, they did so disingenuously, incorrectly attributing contributions to those papers to satisfy the review superficially. Rewarding this behavior with an acceptance sets a troubling precedent for scientific honesty.
>
> Given that the AC bypassed clear procedural guidelines and made favorable assumptions for the authors that were factually incorrect, I am left with serious concerns regarding the objectivity of this decision and the potential for an undisclosed conflict of interest or collaboration between the AC and the authors.
>
> I believe this case warrants a secondary internal review to ensure that the high standards of ICLR are maintained and that the community’s trust in the transparency of the process remains intact.
>
> I am available to discuss the specifics of my technical concerns or the timeline of these events further should you require more detail.
>
> Kind regards,